META-RESEARCH ARTICLE

# The rearing environment persistently modulates mouse phenotypes from the molecular to the behavioural level

Ivana Jaric [1]*, Bernhard Voelkl[1], Melanie Clerc[2], Marc W. Schmid[3], Janja Novak[1], Marianna Rosso[1], Reto Rufener[4], Vanessa Tabea von Kortzfleisch[5], S. Helene Richter[5], Manuela Buettner[6], André Bleich[6], Irmgard Amrein[7], David P. Wolfer[7], Chadi Touma[8], Shinichi Sunagawa[2], Hanno Würbel[1]*

**1** Animal Welfare Division, Vetsuisse Faculty, University of Bern, Bern, Switzerland, **2** Department of Biology, Institute of Microbiology and Swiss Institute of Bioinformatics, ETH Zürich, Zürich, Switzerland, **3** MWSchmid GmbH, Glarus, Switzerland, **4** Department of Oncology-Pathology, Karolinska Institutet, Solna, Sweden, **5** Department of Behavioural Biology, University of Münster, Münster, Germany, **6** Institute for Laboratory Animal Science and Central Animal Facility, Hannover Medical School, Hannover, Germany, **7** Institute of Anatomy, Division of Functional Neuroanatomy, University of Zürich, Zürich, Switzerland; Department of Health Sciences and Technology, ETH Zürich, Zürich, Switzerland, **8** Department of Behavioural Biology, Osnabrück University, Osnabrück, Germany

* ivana.jaric@vetsuisse.unibe.ch (IJ); hanno.wuerbel@vetsuisse.unibe.ch (HW)

**Data Availability Statement:** The ATAC-seq data are available from the NCBI Gene Expression Omnibus (GEO) database under accession number GSE191125. The 16S rRNA gene sequencing data

## Abstract

The phenotype of an organism results from its genotype and the influence of the environment throughout development. Even when using animals of the same genotype, independent studies may test animals of different phenotypes, resulting in poor replicability due to genotype-by-environment interactions. Thus, genetically defined strains of mice may respond differently to experimental treatments depending on their rearing environment. However, the extent of such phenotypic plasticity and its implications for the replicability of research findings have remained unknown. Here, we examined the extent to which common environmental differences between animal facilities modulate the phenotype of genetically homogeneous (inbred) mice. We conducted a comprehensive multicentre study, whereby inbred C57BL/6J mice from a single breeding cohort were allocated to and reared in 5 different animal facilities throughout early life and adolescence, before being transported to a single test laboratory. We found persistent effects of the rearing facility on the composition and heterogeneity of the gut microbial community. These effects were paralleled by persistent differences in body weight and in the behavioural phenotype of the mice. Furthermore, we show that environmental variation among animal facilities is strong enough to influence epigenetic patterns in neurons at the level of chromatin organisation. We detected changes in chromatin organisation in the regulatory regions of genes involved in nucleosome assembly, neuronal differentiation, synaptic plasticity, and regulation of behaviour. Our findings demonstrate that common environmental differences between animal facilities may produce facility-specific phenotypes, from the molecular to the behavioural level. Furthermore, they highlight an important limitation of inferences from single-laboratory studies and thus argue that study designs should take environmental background into account to increase the robustness and replicability of findings.

are available from the European Nucleotide Archive (ENA) under accession number PRJEB49361. All other relevant data supporting the findings of this study are available within the article and its supporting material. The code files for the ATAC-seq analysis are available at: https://github.com/MWSchmid/Jaric-et-al.-2022. The codes used for analysis of behavioural and physiological responses and randomization are available at the Figshare repository https://doi.org/10.6084/m9.figshare.21088642.

**Funding:** This study was supported by the Swiss National Science Foundation (grant 310030_179254) to H.W and core funding from ETH Zürich to the Laboratory of Microbiome Research headed by S.S. The funders had no role in study design, data collection and analysis, decision to publish, or preparation of the manuscript.

**Competing interests:** The authors have declared that no competing interests exist.

**Abbreviations:** ASV, amplicon sequence variant; ATAC-seq, assay for transposase-accessible chromatin using sequencing; *Col19a1*, collagen type XIX α1 chain; DAR, differentially accessible region; *Dlg 2*, discs large homolog 2; EGF, epidermal growth factor; FANS, fluorescence-activated nuclei sorting; FDR, false discovery rate; *Fzd9*, Frizzled9; GEO, Gene Expression Omnibus; GO, Gene Ontology; HPA, hypothalamus–pituitary–adrenal; IVC, individually ventilated cage; KEGG, Kyoto Encyclopedia of Genes and Genomes; LDA, linear discriminant function analysis; LDB, light–dark box; *Lrrc4c*, Leucine-Rich Repeat-Containing 4C; MANOVA, multivariate analysis of variance; OF, open field; PCoA, principal coordinate analysis; PCoA1, principal coordinate axis 1; PERMANOVA, permutational analysis of variance; PND, postnatal day; PSD-93, postsynaptic density protein-93; RF, rearing facility; SRT, stress reactivity test; TGFβ, Transforming growth factor beta; TP, time point; TSS, transcription start site.

## Introduction

The ability to replicate an observation by an independent study is a cornerstone of the scientific method to distinguish robust evidence from anecdote [1]. However, the replicability of original research findings was found to be poor across virtually all disciplines of research [2,3], including preclinical animal research. Thus, the prevalence of irreproducible findings in preclinical research was estimated to be greater than 50% of the published findings [4]. Poor replicability compromises the credibility of animal research, attenuates scientific advances and medical progress, harms animals for inconclusive research, and puts patients in clinical trials at risk. Irreproducibility in biomedical research has mostly been attributed to violations of good research practice, including poor study conduct (e.g., no blinding, no randomisation), small sample sizes resulting in low statistical power, analytical flexibility (including p-hacking and HARKing), selective reporting of findings, and publication bias [2,3,5–11].

In animal research, replicability of experimental findings is complicated by phenotypic plasticity, i.e., the ability of one genotype to exhibit different phenotypes under different environmental conditions [12]. Therefore, phenotypic plasticity increasingly receives attention as an important factor compromising the external validity and replicability of animal research [13–21]. Whereas genotypic differences can be eliminated by selective breeding [22–24], the environment in which laboratory animals are born and grow up may differ substantially between rearing facilities (RFs) [25–27]. As a result, genotype-by-environment interactions throughout ontogeny can lead to phenotypic differences in morphology, physiology, and behavior between animals reared in different environments. Due to such phenotypic plasticity, researchers may fail to replicate research findings, even when using genetically homogeneous (inbred) animals [28–30]. Therefore, phenotypic plasticity may contribute to replication failure and conflicting findings in the scientific literature [13,25,31]. However, the magnitude of this problem is as yet unknown, as existing evidence is generally based on single-laboratory studies [32–34] and experimentally induced environmental interventions. Here, we sought to determine the extent to which common differences in housing and husbandry conditions between RFs modulate the phenotype of the most commonly used inbred strain of laboratory mouse, C57BL/6J. To do so, we used a systematic multicentre approach, whereby C57BL/6J mice from the same breeding stock were allocated to and reared in 5 independent animal facilities, before being transported to a single test laboratory. This approach allowed us to assess the effect of the rearing environment on the phenotype of the mice independent of both genotype (e.g., genetic differences due to genetic drift when mice are obtained from different breeders or different breeding stocks) and test condition (i.e., environmental factors affecting the mice at the time of testing).

Thus, pregnant C57BL/6JRj female mice from a single breeding population (Janvier Labs, Le Genest-Saint-Isle, France) were randomly allocated and transported to 5 independent RFs, where their offspring were born and reared until 8 weeks of age under the facility-specific housing and husbandry conditions (S1A Table). In order to assess the effect of the rearing environment independently of genotype and test conditions, both male and female offspring from all 5 RFs were then transported to a single test laboratory that was new for all mice (S1B Table). Phenotypic differences were assessed at 2 time points (TPs), shortly before shipping the mice from the RF to the test laboratory (TP1) and after 2 weeks of habituation, over the course of a 4-week testing period in the test laboratory (TP2; Figs 1A and S1). Specifically, we examined the extent and persistence of variation in the composition of the gut microbiota associated with the different RFs and measured differences in phenotypic traits such as body weight, adrenal weight, neuroendocrine stress reactivity, and behaviour (Fig 1B). In addition, we assessed differences in neural chromatin accessibility to explore the potential biological

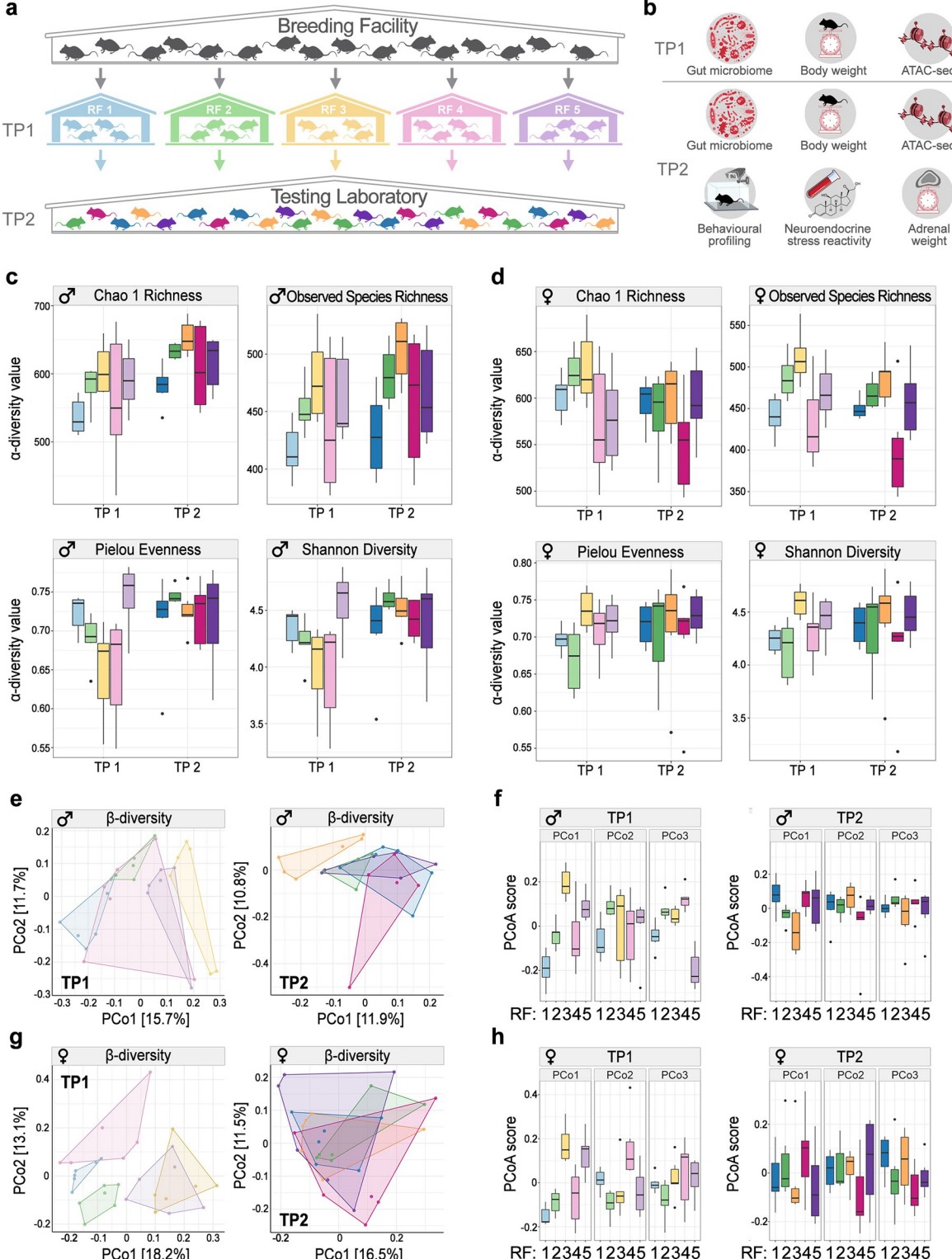

**Fig 1. Study design and effects of RF on gut microbiota diversity and composition. (a)** Schematic illustration of the multicentre study design—genetically homogeneous mice originating from a single inbred stock were reared until the age of 8 weeks in 5 different RFs before testing for phenotypic differences induced by the different rearing conditions in a single testing laboratory. (**b**) Effects of the RF were evaluated at 2 TPs, first, at the end of the rearing period in each of the 5 RFs (TP1) and during the testing period in the testing laboratory (TP2). Outcome measures assessed at both TP1 and TP2 included gut microbiota, body weight, and chromatin profiles using ATAC-seq,

while behavioral tests (open field and light dark box tests) and physiological measures of stress (HPA axis reactivity test (SRT) and relative adrenal weight were limited to TP2). Values for α-diversity metrics for (**c**) male mice and (**d**) female mice from different RFs ($n = 6$ mice/ sex/RF/TP). (**e-g**) Ordination plots visualizing PCoA based on Bray–Curtis dissimilarity between samples of male (**e**) and female (**g**) mice from different RFs, split by TPs. (**f-h**) Differences between loadings of samples on the first 3 PCoA axes in male (**f**) and female (**h**) mice. Box plots show the first and third quartiles; horizontal line represents the median; whiskers represent the mean variability outside the upper and lower quartiles. Individual points represent outliers. TP1: 8 weeks of age (PND 56); TP2: 14.5 weeks of age (PND 104). The raw data underlying this figure are available in the Figshare repository https://doi.org/10.6084/m9.figshare.21082195. The 16S rRNA gene sequencing data are available from the ENA under accession number PRJEB49361. ATAC-seq, assay for transposase-accessible chromatin using sequencing; ENA, European Nucleotide Archive; HPA, hypothalamus–pituitary–adrenal; PCoA, principal coordinate analysis; PND, postnatal day; RF, rearing facility; SRT, stress reactivity test; TP, time point.

basis of behavioural differences (Fig 1B). The study protocol was preregistered (10.17590/ asr.0000201) and is further detailed in the Methods section.

We generated a comprehensive data set by exploring the extent and persistence of variation in the gut microbiome, associated variation in physiological and behavioural traits, and changes in neuronal chromatin organisation as a molecular substrate by which phenotypic differences in behaviour might be mediated. Our study demonstrates that the common environmental differences between RFs can produce facility-specific phenotypes, from the molecular to the behavioural level, thereby compromising replicability of research findings.

## Results

### Rearing facility shaped the gut microbiota composition

The gut microbiome has been reported to play an important role in shaping the host phenotype [35–39]. Therefore, we first examined the extent to which the composition of the gut microbiota varied in response to the macroenvironments of the different RFs, and whether these differences persisted after the transfer to the common macroenvironment of the test laboratory.

We first analysed whether the gut microbiome of mice from different RFs differed in α-diversity measures. In males, effects of the RF were significant for both predicted and observed taxa richness, but there was little effect on overall diversity and evenness (Fig 1C and S2 Table). At TP2, we observed an increase in all metrics of $\alpha$-diversity, except for observed species richness (Fig 1C and S2 Table). Similar patterns in terms of differences between RFs for taxa richness were observed in females; however, there was no change in $\alpha$-diversity metrics across TPs (Fig 1D and S2 Table). Taken together, these results suggest that the macroenvironment of the rRF can lead to significant differences in the richness of the gut microbiome community.

Next, we evaluated the differences in the microbiome composition ($\beta$-diversity) based on the Bray–Curtis dissimilarity between samples [40]. Overall, we found pronounced differences between mice reared in different RFs. When assessing the amount of variation in the data explained by RF at each TP, the effect was most pronounced at TP1, accounting for 28.7% of overall variation in males (Fig 1E and 1F and S3 Table) and 29% in females (Fig 1G and 1H and S3 Table). Importantly, the differences in microbiome composition persisted across TPs, although the amount of variation explained by RF dropped to 20.4% in males and 17% in females. This implies that a large part of the initial differences in the microbiomes of mice from different RFs persisted throughout the 6 weeks in the test laboratory, although they did converge to some extent once they were housed together at the same test facility. When further investigating how overall community composition varied across RFs, we found a clear separation along principal coordinate axis 1 (PCoA1) between RFs 3 and 5 and RFs 1, 2, and 4 in both males (both TPs) and females (TP1). Clustering analysis suggested that the type of mouse diet (specifically diet supplier; S1A Table) was driving the grouping of the mice into these 2 populations (S2A Fig and S1 Data). Interestingly, these 2 populations differed in abundance of

*Firmicutes* and *Bacteroidetes* (S2B Fig), 2 phyla that are associated with numerous phenotypic differences in health and disease in animal and human studies [41–43]. Overall, these findings show that the rearing environment can lead to substantial and temporally persistent compositional differences in the gut microbiome community, which, in turn, may drive phenotypic variation [38].

## Rearing facility persistently affected body weight

To assess phenotypic differences between mice from different RFs, we first measured the body weight of the mice, which often correlates with other physiological variables and can potentially serve as latent variable in studies on mouse energy metabolism and metabolic diseases [44–46]. Although body weight is a highly heritable trait, we found that genetically homogeneous mice reared in different RFs differed markedly in body weight, and these differences persisted at the test laboratory throughout the experiment. RF was the only factor that had a strong and persistent effect on body weight in both males and females, while variation in litter size, litter sex ratio, and group size after weaning had no significant effects (Fig 2A and S4 Table). This finding is in line with the recent findings of Corrigan and colleagues [44], who found that the institution where experiments were conducted was among the main factors influencing metabolic rate and body weight in C57BL/6J mice.

## Rearing facility modulated the behavioural phenotype

To analyse phenotypic differences in behaviour, we combined variables derived from 2 standard behavioural tests (open field (OF) and light–dark box (LDB)) in a multivariate analysis of variance (MANOVA). We found that RF had a strong effect on the behavioural phenotype, explaining 14.4% and 12.4% of the total variation in males and females, respectively. After controlling for fixed effects, RF even explained 23% and 25% of the remaining variation in behavioral outcomes (partial η2 estimate). Moreover, whereas variation in litter size, litter sex ratio, and group size after weaning had no effect on behaviour in both males and females, oestrous cycle stage on the test day had a strong effect in females (S5 Table and S3A–S3C Fig).

Using linear discriminant function analysis (LDA) on the combined behavioural data (S6 Table), we were able to correctly classify 58% of all male mice and 53% of all female mice according to their RF, which is substantially more than the 20% predicted by chance ($X^2$ = 55.1, $p = 1.14 \times 10^{-13}$ and $X^2$ = 41.7, $p = 1.08 \times 10^{-10}$, respectively, for males and females; Fig 2B and S7 Table). In males, the first 2 discriminant functions together explained 79% of the variation between RFs, whereby the coefficients of the discriminant functions indicate that distance travelled in the OF and time in the light compartment in the LDB, 2 main measures of exploration and emotionality, contributed most to the first function, while time in the light compartment and number of entries into the light compartment in the LDB contributed most to the second function. In females, the first 2 functions together explained even 90% of the between-facility variance. Distance travelled in the OF contributed most to the first function, while time in the centre in the OF contributed most to the second function. These findings demonstrate that common environmental differences between animal facilities during the first 8 weeks of postnatal development can substantially alter key aspects of the behavioural phenotype of mice, which persist into adulthood.

## Rearing facility did not affect neuroendocrine stress reactivity but adrenal weight

Further, we predicted that differences in the environmental conditions between RFs during the late prenatal, early postnatal, and adolescent period may differentially shape the reactivity

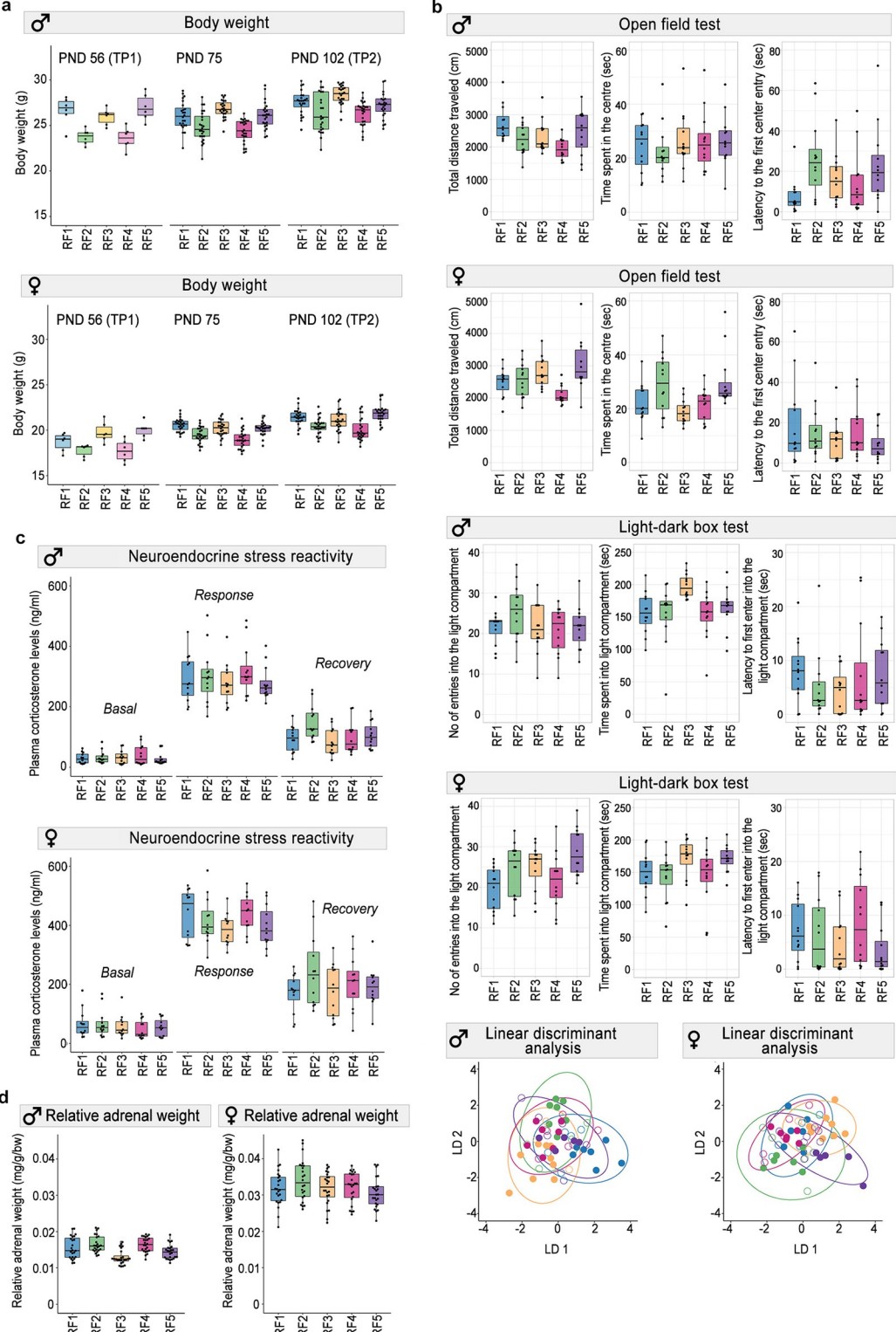

**Fig 2. Effects of RF on the behavioral and physiological profile of the mice. (a)** Body weight persistently varied by RF in both males and females (*n* = 6 mice/sex/RF for TP1; *n* = 24 mice/sex/RF for TP2). **(b)** Behavior of the mice varied consistently by RF both in males and females. In the LDA plots, color indicates RF, and the circles represent classification based on discriminant function analysis (*n* = 12 mice/sex/RF). **(c)** RF did not affect plasma corticosterone levels in the SRT both in males and females (*n* = 12/mice/sex/RF), while relative adrenal gland weights (*n* = 24/mice/sex/RF) were affected

only in males (**d**). Box plots include individual data points and show the first and third quartiles; horizontal line is the median; whiskers represent the variability outside the upper and lower quartiles. TP1: 8 weeks of age (PND 56); TP2: 14.5 weeks of age (PND 104). The raw data underlying this figure are available in the Figshare repository https://doi.org/10. 6084/m9.figshare.21081949. LDA, linear discriminant function analysis; PND, postnatal day; RF, rearing facility; SRT, stress reactivity test.

of neuroendocrine systems, including the reactivity of the hypothalamus–pituitary–adrenal (HPA) axis to stressors later in life [47–50]. We, therefore, examined whether RF altered the animals' HPA stress reactivity by measuring changes in plasma corticosterone during and after a brief period (20 minutes) of physical restraint in a plastic tube. However, we found no consistent differences in HPA stress reactivity between mice reared in different RFs (Fig 2C and S8 Table). In males, there was a strong effect of litter size on basal corticosterone levels, and group size after weaning strongly affected both basal levels and acute response levels of corticosterone (S8 Table). As expected, corticosterone levels in females were almost double those in males [51,52] (Fig 2C), and the oestrous cycle stage had a strong effect on basal corti- costerone levels (S8 Table and S4 Fig). However, we found that RF had a strong effect on adre- nal weight, at least in males (Fig 2D and S9 Table). These results suggest that the chronic stress engendered by standard housing conditions and husbandry procedures induced changes in adrenal gland morphology and function, which may have buffered the neuroendocrine stress response to acute stressors [53].

## Rearing facility influenced chromatin organisation in neuronal nuclei

We next explored at the molecular level whether environmental differences between RFs affect epigenetic mechanisms in brain areas involved in behavioural control. More specifically, we assessed chromatin plasticity since chromatin is considered as an interface between the envi- ronment and the genome and plays a key role in the regulation of gene expression, mediating various aspects of plastic behavioural responses to environmental perturbation [54,55]. Since chromatin organisation is tissue and cell type specific, we performed the assay for transposase- accessible chromatin using sequencing (ATAC-seq; [56]) on neuronal nuclei extracted from the ventral hippocampus, a brain area involved in modulating emotional behaviour and stress responses in mice [57,58]. This analysis was limited to males, as they showed more pro- nounced phenotypic variation, especially in behavioural traits.

We found that most samples clustered based on RF, indicating pronounced differences in chromatin accessibility. This pattern was observed by looking at both the overall dissimilarity of all ATAC-seq profiles and the 10% most variable peaks (Fig 3A). RF explained 55.33% and 36.79% of overall variation at TP1 and TP2, respectively (Figs 3B and S5 and S10 Table). Remarkably, variation explained by RF was much larger in the open chromatin sites (77.5% at TP1 and 70.9% at TP2) than in the closed sites (48.2% at TP1 and 28.4% at TP2), suggesting that these differences have functional consequences (Fig 3B and S10 Table).

In terms of genomic features, the most accessible sites were preferentially located within the promoter regions, further corroborating the potential functional significance of the observed changes, while the less accessible sites were mainly located in the intergenic regions and introns (Fig 3C and 3E). In addition, the number of genes associated to the most accessible peaks were common between subjects, whereas less accessible sites and associated genes behaved much more randomly and decreased with the number of selected samples (Fig 3D).

Next, we generated lists of all differentially accessible regions (DARs) in the ventral hippo- campus of mice from different RFs and mapped them to their adjacent genes for all compari- sons at both TPs separately (S2 Data). Among the genes with the highest fold change, we, for

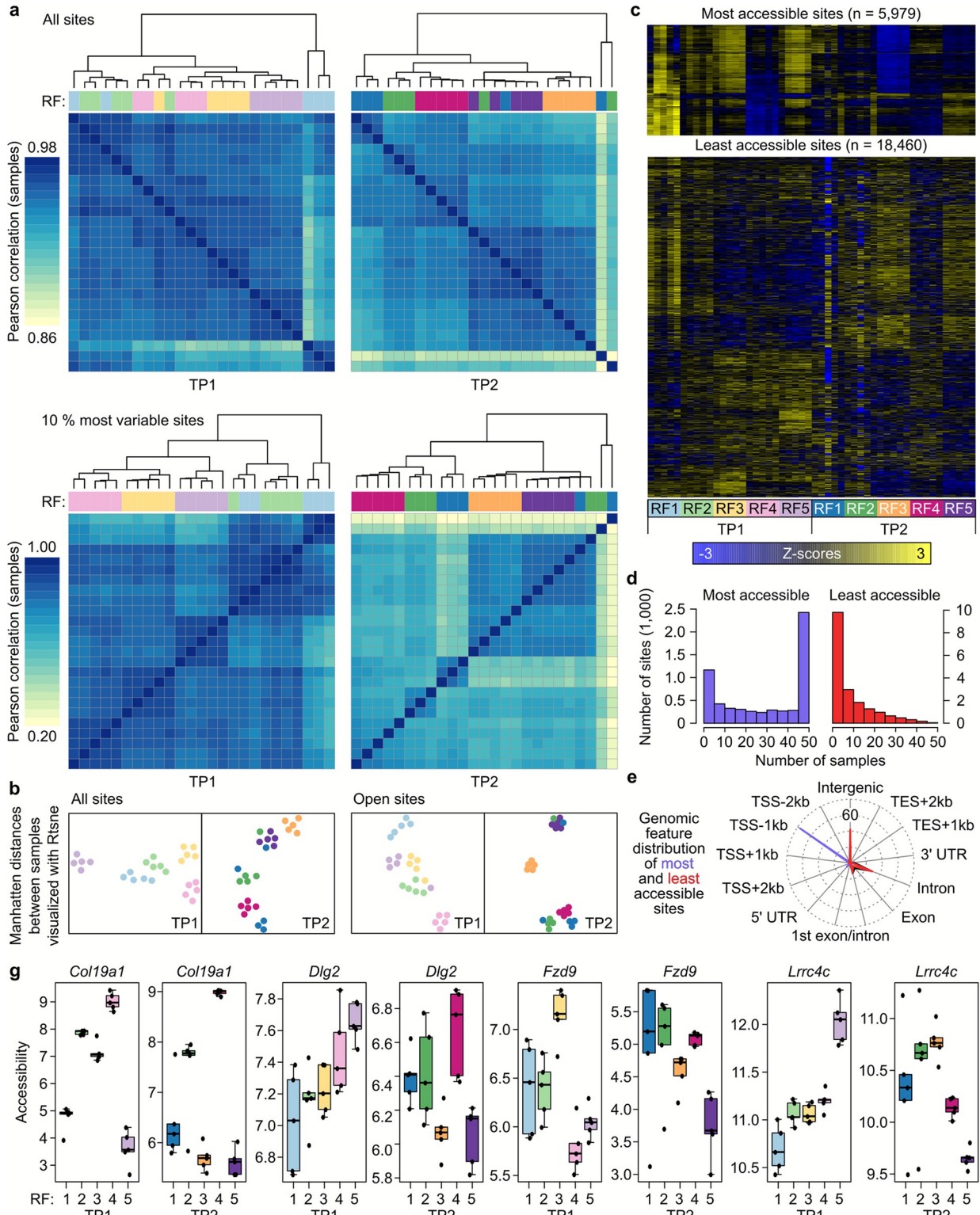

**Fig 3. Differences in neuronal chromatin accessibility between males from different RFs. (a)** Sample correlation matrices based on all sites and the 10% most variable ATAC-seq sites (*n* = 5 mice/RF/TP). (**b**) Manhatten distances between samples for all sites and open chromatin sites visualised by t-SNE. (**c**) ATAC-seq accessibility signal in response to the different RFs and TPs. Heatmap representation of the most and least accessible sites. The color represents the intensity of chromatin accessibility, from gain (yellow) to loss (dark blue), calculated by using row wise Z-scores (the values are scaled by subtracting the average across samples and by dividing by the standard deviation across samples). (**d**) Bar

graphs representing the number of genes associated with the most and least accessible peaks. (**e**) Spidergraph representing the genomic features mapped by all (black), open (blue), or closed (red) sites. (**f**) Chromatin accessibility profiles of *Col19a1*, *Dlg 2*, *Fzd9*, and *Lrrc4c*. The raw data underlying this figure are available from the NCBI GEO database under accession number GSE191125. The *analysis script is available at* the GitHub repository https://github.com/MWSchmid/Jaric-et-al.-2022. ATAC-seq, assay for transposase-accessible chromatin using sequencing; GEO, Gene Expression Omnibus; RF, rearing facility; TP, time point.

instance, found *Col19a1* (encoding collagen type XIX α1 chain), where chromatin accessibility differed between RFs and remained consistent across both TPs (Figs 3G and S5E). This indicates that persistent chromatin regulation occurred during the rearing period in response to the specific environment of the RF in genes necessary for hippocampal synapse formation [59–61]. We also found DARs between RFs that changed between TP1 and TP2 and mapped them to genes such as *Dlg 2* (discs large homolog 2, also known as postsynaptic density protein-93 (PSD-93), *Fzd9* (encoding Frizzled9, one of the Wnt receptors), and *Lrrc4c* (encoding Leucine-Rich Repeat-Containing 4C). Changes in these genes, important for postsynaptic plasticity (Figs 3G and S5E) [62–65], indicate that mice from different RFs were using different chromatin regulation strategies to adapt to the new environment of the test laboratory.

### Rearing facility induced chromatin changes relevant to neuronal function

To assess the functional significance of environmentally induced chromatin changes, we performed Gene Ontology (GO) and Kyoto Encyclopedia of Genes and Genomes (KEGG) pathway analyses, which focused on genes mapped to DARs located near transcription start site (TSS).

The GO analysis revealed that differences between RFs persistently influenced nucleosome function and regulatory processes important for hippocampal synaptic plasticity and neurogenesis, such as the response to epidermal growth factor (EGF) [66], regulation of Notch [67–69], and Transforming growth factor beta (TGFβ) receptor signaling [70,71] (Fig 4A and 4B). There was also a clear effect of RF on genes involved in the regulation of various behavioural processes and presynaptic plasticity events, targeting mainly GABAergic and glutamatergic transmission (S3A and S3C Data). This effect was evident only at TP1, while at TP2, enriched terms were associated with the modification of postsynaptic structure, regulation of actin cytoskeleton, dendrite development, and neurotransmitter receptor complex (S4B and S4D Data and Fig 4A and 4B).

The KEGG analysis highlighted an overrepresentation of genes belonging to adherens junction, dopaminergic synapses, hippo, apelin, and insulin resistance signaling pathway. These pathways, which have an important role in maintaining hippocampal development, morphology, and plasticity, were significantly affected by the RF at both TPs and likely have functional consequences for behavioural regulation [72–75]. Enrichment of genes relevant to neurotrophin, long-term depression and potentiation, serotonergic and glutamatergic synapse pathways was evident only at TP1 (Figs 4C and S4A). These differences diminished after the mice had spent 6 weeks in the test laboratory. At TP2, we noticed significant differences for Notch, prolactin, relaxin, and AMPK signaling pathways, indicating their potential role in behavioral adaption to the new environment (Figs 4C and S4A).

Overall, these findings demonstrate that facility-specific macroenvironments influenced developmental programs during the late prenatal, early postnatal, and adolescent period, by affecting neuronal chromatin accessibility profiles and shaping the mice's behavioural phenotypes.

## Discussion

In this study, we found that common differences in standard housing and husbandry practices between animal facilities modulated morphological, physiological, and behavioural traits in a

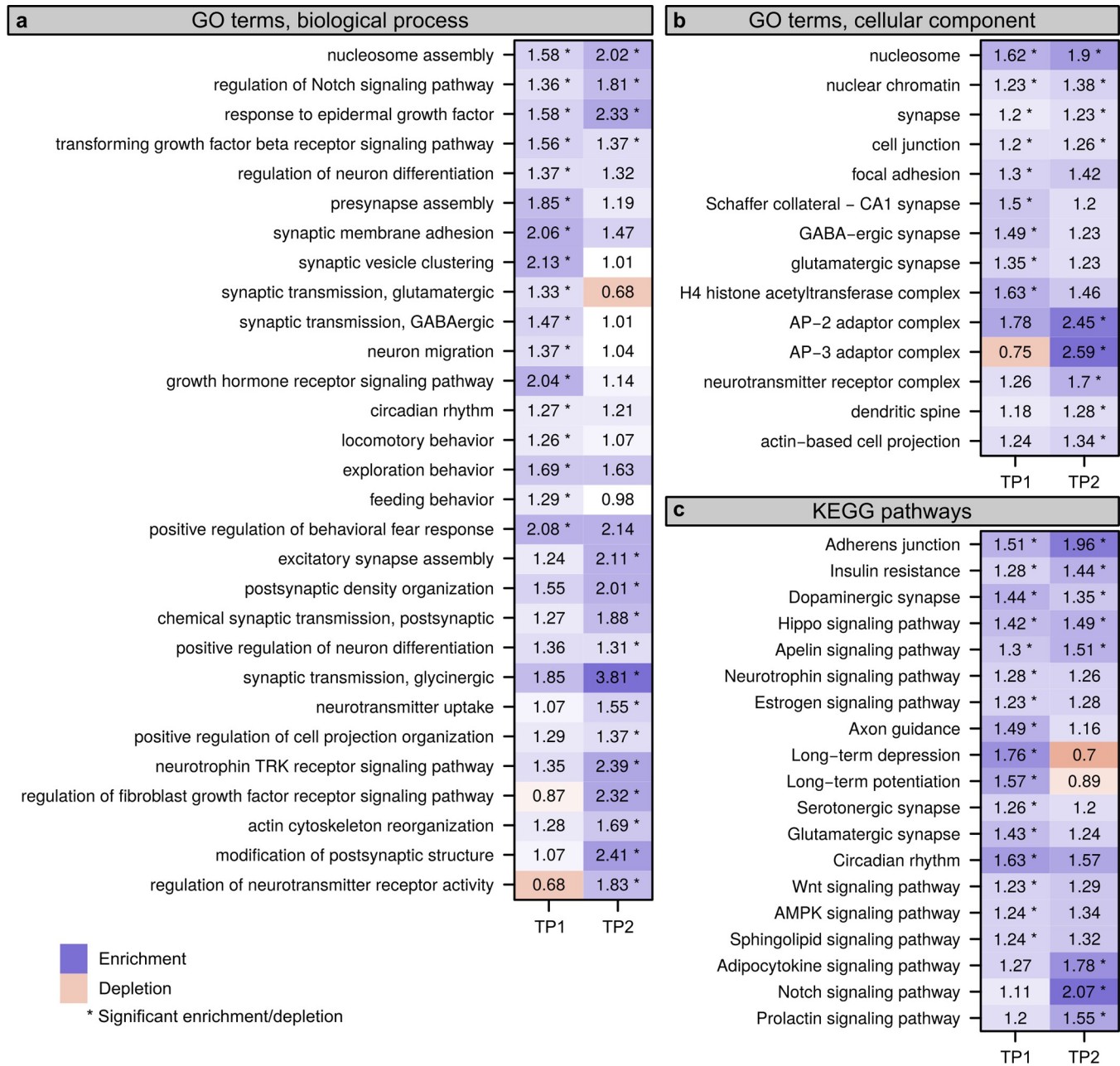

**Fig 4. Predicted biological processes, cellular components, and KEGG pathways affected by differential rearing environment.** GO analysis of genes showing differential chromatin accessibility between the different RF presented for each TP (**a-b**); KEGG pathway analysis of the genes showing differential chromatin accessibility between the different RFs for each TP. Fields marked with an asterisk depict comparisons that were statistically significant between TPs (two-sided Fisher's exact test, adjusted for multiple testing, FDR < 0.05). FDR, false discovery rate; GO, Gene Ontology; KEGG, Kyoto Encyclopedia of Genes and Genomes; RF, rearing facility; TP, time point.

genetically homogeneous (inbred) cohort of C57BL/6J mice from a single breeding colony, thereby producing facility-specific phenotypes. Phenotypic plasticity is ubiquitous in nature, and studies under controlled laboratory conditions have long shown that many aspects of housing and husbandry conditions can alter the phenotype of mice throughout ontogeny [76–83]. In fact, this is the main reason why textbooks of laboratory animal science recommend

environmental standardisation to minimise variation in experimental results [22]. However, previous studies have shown that such standardised experiments characterizing the phenotype of mice may produce results that are idiosyncratic to the laboratory where the study was conducted [44,84]. Our findings corroborate and extend these findings. In previous multicentre studies [44,84], the effect of the rearing environment was confounded with the specific test conditions at the test centre. Thus, in these studies, phenotypic differences between animals tested in different facilities may reflect differences in the state of the animals while being tested. In contrast, in the present study, phenotypic differences between mice from different RFs must reflect differences in the underlying phenotypic traits. This is guaranteed by the fact that all mice were derived from a single breeding colony (minimizing the risk of genetic differences between mice from different RFs), that they were all tested in the same test laboratory under the same test conditions by the same experimenters, and that test order was counterbalanced for RF.

We also confirmed and extended previous findings demonstrating that variation in microbiome composition is strongly determined by the rearing site [37,85–88]. Previous studies found that although vendor-specific differences in the gut microbiome of C57BL/6J mice may decrease over time at a new site, they persisted throughout studies lasting for 12 weeks [37] and 24 weeks [27], respectively. However, these mice had been housed in individually ventilated cages (IVCs) or microisolater cages, preventing contamination between cages. Our results extend these findings by showing that such animal facility-dependent differences in microbiome composition persisted in the testing facility, even though our mice were housed together in 1 housing room in conventional open cages.

Further, our findings suggest that environmental differences between RFs influence neuronal developmental patterns by modulating gene regulatory networks involved in the regulation of hippocampal synaptic plasticity and neurogenesis. Such effects on the chromatin profile of functionally relevant genes may be responsible for persistent changes in behavioural traits [89,90]. However, we also found that some of the pathways affected by the rearing environment maintained plasticity, possibly to facilitate adaptation to environmental change, as shown by chromatin reorganisation in response to the transfer to the test laboratory at 8 weeks of age. This result deserves further investigation as chromatin plasticity is considered to provide a molecular mechanism for adaptive plasticity under different environmental conditions, as shown for example in *Drosophila melanogaster* [91].

Our findings raise important questions about the mechanism underlying the observed phenotypic differences that cannot be answered based on the present data but need to be tested independently in follow-up studies. For example, given the observed association between diet and microbiome composition, the question arises whether such differences would also occur if different diets were fed in the same facility, and whether diet might also explain some of the variation in epigenetic and behavioural profiles. Several studies have shown that gut microbiota can affect brain development and behavioral functions [92,93]. These effects might be mediated by gut-microbiota products that can cross the blood brain barrier and modulate the chromatin landscape in neurons, which, in turn, can influence behavior [94–96]. Controlled experiments with different diets, fecal transplants, and treatment with antibiotics would be needed to elucidate these potential causal mechanisms further.

In conclusion, our findings could help to explain replicability issues in animal research [2,97]. Poor replicability has mostly been attributed to publication bias, lack of statistical power, analytical flexibility, and other risks of bias [3,7,98–100], albeit empirical evidence has remained elusive [101]. In contrast, the large between-study heterogeneity caused by rigorous within-study standardisation has long been ignored as a cause of poor replicability, despite both theoretical and empirical evidence [13,14,25,31,83,102–106]. Our findings highlight an

important limitation of inferences from single-laboratory studies and suggest that the (early) environmental background of animals—just like their genetic background [107–109]—should be accounted for by study design to produce more robust and replicable research findings. Thus, results from standardised single-laboratory studies might only be considered as preliminary evidence. Whether using animals from multiple breeding or RFs provides an effective solution to systematically heterogenise the environmental background of study populations remains to be tested. However, we hope that our findings will stimulate research to find other, perhaps more practicable ways to produce robust and replicable research results.

## Methods

### Study preregistration and ethical statement

Before data acquisition started, the study protocol was preregistered under the DOI number: 10.17590/asr.0000201 at the Animal Study Registry, operated by the German Centre for the Protection of Laboratory Animals (Bf3R) at the German Federal Institute for Risk Assessment (BfR).

All animal experiments were conducted in full compliance with the Swiss Animal Welfare Ordinance (TSchV 455.1) and were approved by the Cantonal Veterinary Office in Bern, Switzerland (permit number: BE12/19). RFs located in Hannover and Münster did not require separate approval by governmental authorities for an experimental animal study, because animals were only housed in those laboratories, while all experimental procedures were carried out in Bern under the abovementioned license.

### Animal subjects and study design

Nine-week-old, time-mated, primiparous pregnant C57BL/6JRj females in the last third of pregnancy (gestational day 14 or 15), all derived from the same breeding stock and colony room of a commercial breeder (Janvier Labs, Le Genest-Saint-Isle, France), were randomly allocated to 5 different rearing animal facilities ($n$ = 18 per facility). The RFs were located at the following institutions: (**i**) Institute of Laboratory Animal Science, Hannover Medical School, Germany (RF 1 and RF2); (**ii**) Division of Animal Welfare, Vetsuisse Faculty, University of Bern, Switzerland (RF 3); (**iii**) Department of Behavioural Biology, University of Münster, Germany (RF 4); and (**iv**) Institute of Anatomy, University of Zürich, Switzerland (RF 5).

Pregnant dams were singly housed for approximately 5 days, from arrival at the RF until parturition. Dams were monitored daily for parturition and day of birth was defined as postnatal day 0 (PND 0). Litters were not culled during the lactation period and all healthy pups were weaned at PND 22 according to predefined weaning criteria (S6 Fig). At weaning, in each RF from the 18 possible litters, all litters that contained at least 3 male and/or female pups were used to randomly select 12 groups of 3 littermates for each sex. If fewer than 12 groups with at least 3 littermates of a sex were available, these were complemented by litters with at least 2 pups of that sex. This strategy resulted in a total of 326 weaned mice. Litter size, litter sex ratio, as well as the final number of weaned males and females from each litter are presented in S11 Table.

From each litter, 3 (or 2) pups per sex were selected randomly and reared together until the age of 8 weeks (PND 56) according to the specific protocols of housing and husbandry of each of the 5 RFs (e.g., type of cages, handling method, bedding, nesting material, diet, light regime). Detailed housing and husbandry conditions are reported in S1A Table. The first 8 weeks of postnatal life were selected because they cover 2 sensitive developmental periods (i.e., early life and adolescence) in mice [110]. These periods represent a critical stage of brain [111–115], HPA axis [48,116], and gut microbiome development [39,117] when environmental inputs may shape the later-life phenotype of mice at different levels of organisation.

The effect of rearing environment was evaluated at 2 TPs (Figs 2B and S1). At TP1, 1 mouse per sex and cage of all cages with 3 mice was weighed and killed within each of the 5 RFs and brain and cecal samples were harvested for epigenomic and microbiome analyses. To ensure that observed differences are attributed only to the differential rearing environments, we standardised the tissue collection procedure. All mice were killed at the beginning of the light cycle (within the first 3 hours after the lights were turned on) by the same person who was not in contact with those animals before and who was using the same equipment in each of the 5 RFs. Blinding with regard to RF was not possible for weighing and organ collection since the experimenter needed to travel to each RF.

At PND 58, the remaining pairs of male and female offspring ($n$ = 240; 24 mice per sex per RF) were transferred from the 5 RFs to the testing laboratory, located at the Vetsuisse Faculty of the University of Bern (testing laboratory). The testing lab in Bern was separate from the RF in Bern. To ensure that all animals experienced approximately the same treatment during transport to the testing facility, the driver of the car transporting the animals from the RF in Bern to the adjacent test laboratory in Bern was asked to take a 2-hour detour. Special efforts were made to reduce any possible transport-induced stress. The same-sex cage mates were placed into 4 compartment transport boxes (Type 500005L; Bio Services BV, Uden, the Netherlands), with 2 mice per compartment and shipped by a professional company using an environmentally controlled vehicle. Each compartment contained 1 cm of bedding, nesting material from the home cage, food pellets, and hydrogel.

Upon arrival, animals were checked for health and pair housed in freshly bedded Type 3 cages (floor area 820 cm$^2$) and habituated to the new animal facility for 2.5 weeks. Each Type 3 cage contained 3 cm of bedding (OSafe Premium Bedding, SAFE FS 14, Safe-Lab, Rosenberg, Germany), a red mouse house (Tecniplast, Indulab, Gams, Switzerland), a medium-size cardboard tunnel (Play tunnel, #CPTUN00016P, Plexx B.V., the Netherlands), and 10 g of nesting material (Sizzle Nest #SIZNEST00016P, Plexx B.V., the Netherlands). Standard rodent chow (Kliba Nafag #3430, Switzerland) and tap water were available ad libitum. Females and males were housed in separate rooms, and all animals were kept on a 12:12 light/dark cycle, with lights on at 12:00 h. Detailed housing and husbandry conditions in the testing facility are reported in S1B Table.

The day after arrival in the testing facility, animals were individually marked by ear tattoo, after which cages were assigned new identification numbers, and positions of cages within and between cage racks were randomly reshuffled as part of the blinding procedure. Further information on blinding is available as a supporting text.

Behavioural and physiological phenotyping commenced after an acclimatisation period of 2.5 weeks. We focused on phenotypic traits of exploration, emotionality, and stress reactivity that are known to be sensitive to environmental variation during early ontogeny [48,112,118,119]. Two common tests for exploration and emotionality, the OF test and the LDB test, were conducted in that order, with a break of 7 days in between, followed by an SRT after another break of 7 days. For these tests, 1 mouse per cage, sex, and RF ($n$ = 120) was used. After 1 additional week (around 14.5 weeks of age; PND 102; TP2), all mice ($n$ = 240) were killed for postmortem analyses. Body weights were also recorded for each mouse during cage changes throughout the acclimatisation period and prior to killing. To avoid possible influences of the circadian rhythm on behaviour, corticosterone secretion, and molecular readouts, all procedures were performed during the light phase (from 12:00 h to 16:30 h).

## Tissue sampling procedure

All animals were killed by cervical dislocation. Whole brains were immediately removed and quickly frozen in a hexane bath on dry ice before being stored at −80˚C. The brain region of

interest, i.e., ventral hippocampus, was dissected later for subsequent molecular analyses. Adrenal glands were removed, dissected from fat, and weighed using a precision scale (Mettler AE160, Mettler-Toledo, Switzerland). The mouse cecum together with its content was isolated, snap frozen in liquid nitrogen, and kept at −80°C for microbial DNA extraction and sequencing.

## Gut microbiota composition analysis

For gut microbiota analysis, samples of mouse caeca were taken at both TPs, at each RF, and at the end of the experiment at the testing laboratory. Six cages per sex and RF were randomly selected from all cages containing 3 littermates after weaning ($n$ = 180). One mouse from each selected cage was killed in the RF at 8 weeks of age (TP1; PND 56; $n$ = 60), while the other 2 cage mates ($n$ = 120; one underwent behavioural testing and one remained naïve) were killed in the testing laboratory at TP2 (14.5 weeks of age; PND 102).

Microbial DNA was extracted from cecal content using the Allprep DNA/RNA mini kit (Qiagen, Hilden, Germany) according to the manufacturer's instruction. DNA was eluted and concentrations were assessed using the QuantiFluor dsDNA system (Promega).

The 16S rRNA V4 region was amplified using the following primers 515f-Y: GTGY-CAGCMGCCGCGGTA and 806r-N: GGACTACNVGGGTWTCTAAT and The Q5 high-fidelity DNA polymerase kit (New England BioLabs, UK). In a total volume of 25 µL, 2 µl of extracted DNA was added to a PCR reaction mix prepared by mixing a final concentration of 1X Q5 reaction buffer, 200 µM of dNTPs, 0.5 µM of each primer, 0.02 U/µL of Q5 5 High-Fidelity DNA Polymerase, and 1X of Q5 High GC enhancer. The first PCR reaction was carried out using the following conditions: (i) first denaturation: 95°C for 30 s; (ii) denaturation in each PCR cycle: 98°C for 10 s; (iii) annealing: 56°C for 30 s; (iv) extension: 72°C for 30 s; (v) final extension at the end of the reaction: 72°C for 2 min, followed by a hold step at 4°C. The cycles 2 to 4 were repeated 8 times. The PCR products were purified using CleanNA CleanNGS purification beads (CNGS0050; LabGene Scientific SA), resuspended in 15 µl of EB buffer, and served as templates in the second PCR reaction. In the second PCR step, unique dual index barcodes of length $2 \times 8$ nt were added to each sample, which allowed equimolar pooling of samples after quantification of the target product using the Agilent fragment analyzer (Agilent). In total, the final library pool contained 176 samples, 3 bacterial mock communities, and 20 DNA extraction blanks. The finished library pool was sequenced using the NovaSeq 6000 platform (Illumina, USA) in a single lane of SP flow cell at the Functional Genomics Center Zürich.

The raw sequencing data were analysed using the DADA2 pipeline (version 1.14 16), and individual reads were grouped into amplicon sequence variants (ASVs). The final table contained 1,560 ASVs. After removal of the blank and mock samples from the data, the individual library sizes ranged from 47,910 to 1,616,753, with a median of 843,361 (S6A Fig). To mitigate the effect of variation in library size across samples, we performed random down-sampling of reads within each sample to an even library size across samples. Given the minimum read count in the data, counts were rarefied to a depth of 47,000 reads per sample (S6B Fig). We calculated both α diversity (within sample diversity) and β diversity (between-sample diversity) in order to assess the effect of RF on microbial community composition within and between mice.

For α diversity, the following metrics were calculated: (i) observed species richness, which represents the total number of species counted within a sample; (ii) Chao1 richness, for estimation of the "true" species diversity, which is calculated using the following formula: $Chao_1 = S_{obs} + \frac{F_1^2}{2F_2}$, where $S_{obs}$ stands for the observed number of species, and $F_1$ and $F_2$ stand for the number of species with 1 or 2 observed reads, respectively; (iii) Shannon diversity

index, which illustrates the diversity within a sample, taking both richness and evenness into account. It is calculated using the following formula: $H^{t'} = -\sum_{i=1}^{s} p_i ln(p_i)$, where $S$ represents the total number of species and $p$ represents the proportion (n/N) of individuals of 1 particular species found (n), divided by the total number of individuals found (N); and (iv) Pielou evenness for estimation of how similar in numbers each species in a sample is. It is calculated using the formula $J = \frac{H^{t'}}{ln(S)}$, where $H^{t'}$ is the Shannon index and $S$ is the total number of species in a sample. ANOVA was used to test the effect of RF and TP on α-diversity measures.

To analyse changes in microbiota composition between mice reared at different TPs or RFs, the Bray–Curtis dissimilarity between samples was calculated using the formula $BC_{ij} = 1 - \frac{2C_{ij}}{S_i + S_j}$, where $i$ and $j$ are the 2 samples, $S_i$ is the total number of species counted in sample $i$, $S_j$ is the total number of species counted in sample site $j$, and $C_{ij}$ is the sum of only the lesser counts for each species found in both samples. Principal coordinate analysis (PCoA) was used to visualise the similarity in microbiome composition between samples and to retrieve sample loadings onto the first 3 PCoA axes. A permutational analysis of variance (PERMANOVA) was used to assess the proportion of variation in microbiome composition explained by RF and age (i.e., TP).

Based on our study design, there were 3 groups of cecal samples: (i) samples collected at TP1 (within each of RFs); (ii) samples collected at TP2 from mice that underwent behavioural testing; and (iii) samples collected at TP2 from mice that did not undergo behavioural testing and that were used for chromatin profiling. To balance the study design (there were double number of samples collected at TP2 comparing to the number of samples collected at TP1), we assessed whether the 2 sets of samples (from behaviourally tested BT mice and mice used for molecular analysis, i.e., chromatin profiling MA mice) collected at TP2 differ in terms of species richness, diversity, and overall community composition. Our analysis showed that there were no statistically significant differences in either α diversity (assessed using Wilcoxon signed-rank test) or β diversity (assessed using PERMANOVA) in any of the tested parameters between BT and MA mice (S7 Fig). Therefore, only MA mice were considered for the final analyses. Differential abundance analysis was performed by using an adjusted *p*-value threshold of 0.05 and a log2-fold change threshold of 1.

All analyses related to the gut microbiome were done in R (version 4.1.0) using the libraries vegan (2.5–7) for down-sampling, PERMANOVA, and calculation of observed/chao1 species richness, microbiome (1.14.0) for calculation of Shannon diversity and Pielou evenness, ampvis2 (2.7.2) for PCoA ordination, stats (4.1.0) for ANOVA calculations and hierarchical clustering, fpc (2.2–9) for definition of cluster number, factoextra (1.0.7) and ggdendro (0.1.22) for dendrogram creation, DESeq2 (1.32.0) for differential abundance analysis, and ggplot2 (2.2.1) and patchwork (1.1.1) for visualisation.

## Testing for behavioural and physiological responses

The set of behavioural outcomes belongs to the confirmatory part of the study, and appropriate sample size was determined a priori by a power analysis using simulated sampling for a two-way ANOVA design. The power analysis was done for the main outcome variable, plasma corticosterone levels in the SRT. Based on historical data [120,121], we expected to observe an effect of medium size (i.e., means estimates for 2 randomly chosen RFs are expected to be in the range of 20%, equivalent to a ratio of between-facility: within-facility variation of 1:2). This resulted in a required minimal sample size of 12 mice per RF and sex. Further information on the sample size calculation is available in the supporting information file (S8 Fig). The same sample size (*n* = 12 per RF and sex) was also used for behavioural testing.

The order of behavioural test trials for all mice was randomised using the random number generator of the Mathematica software (version 11; Wolfram Research, Champaign, Illinois, USA). Forty mice (20 males and 20 females) were randomly assigned to each of 3 experimenters. Mice were handled by the same experimenter during the habituation period, cage change, and behavioural testing. Animals were tested in parallel (at the same time, but in separate apparatuses) by 2 experimenters, with 3 different combinations of 2 experimenters each day. Testing was carried out in batches during 4 consecutive days. Males were tested on the first and third day, while females were tested on the second and fourth day. Each day's testing was done by each experimenter in 3 blocks of 5 animals, each. The randomisation and allocation procedures were restricted so that, in each block for each experimenter, there was exactly 1 mouse from each RF in random order, with the addition that in no case animals tested at the same time were from the same RF.

## Open field test

The OF test was performed in a polycarbonate box (45 × 45 × 45 cm; illumination set to 120 lux) with grey walls and a white base plate. Each mouse was placed in the left (close to the experimenter) corner, facing the wall, and allowed to freely explore the open field for 10 min. Recording started immediately after placing the animal in the box. The behaviour of the first 5 min of the test was analysed.

The behaviour was video recorded using an infrared camera system, and mice were automatically tracked from videos using EthoVision XT software (version 11.5; Noldus, Wageningen, the Netherlands). The space was virtually divided into a centre zone (20 × 20 cm) and an outer zone. The total distance traveled, the average velocity, the number of entries into the centre area, the time spent in the centre, and the latency to the first centre entry were scored.

## Light–dark box test

The LDB test was conducted in a box (37.5 × 21.5 × 15 cm) consisting of a small, closed dark compartment (12.5 × 21.5 × 15 cm; illumination set to 5 lux) and a larger light compartment (25 × 21.5 × 15 cm; illumination set to 200 lux) connected by a sliding door. Each mouse was placed in the dark compartment of the apparatus and testing began after the delay of 5 s as the sliding door to the light side of the box was raised, and the duration of the test was 10 min. The behaviour of the first 5 min of the test was analysed. The total distance traveled, the average velocity in the light compartment, the time spent in the light compartment, the number of entries into the light compartment, and the latency to enter the light compartment were measured from video recordings using EthoVision XT software (version 11.5; Noldus, Wageningen, the Netherlands).

## Stress reactivity test

The SRT was performed according to established protocols [122] with slight modifications. In brief, each mouse was taken out of its home cage and a first blood sample was collected by incision of the ventral tail vessel with a scalpel blade (Paragondisposable sterile scalpels No. 10, Paragon Medical, Lausanne, Switzerland). The procedure was limited to 2 min to obtain basal levels of corticosterone unaffected by the sampling procedure. Immediately after blood collection, the mouse was restrained for 20 min in a 50-ml plastic conical tube (11.5 cm × 2.5 cm; Fisherbrand Easy Reader, Fisher Scientific AG, Reinach, Switzerland) with custom-made holes for breathing and for the tail. At the end of the 20-min restraint period, a second blood sample was taken from a fresh incision rostral to the first one, followed by placing the mouse back in its home cage. A third blood sample was taken from a third incision rostral to the second one,

90 min after the onset of restraint. Each blood sample was collected by using a dipotassium-EDTA capillary blood collection system (Microvette CB 300 K2E, Sarstedt, Nümbrecht, Germany). Immediately after sampling, the blood samples were placed on ice. Within 60 min, the samples were centrifuged for 10 min at 4,000g and 4˚C. Plasma samples were transferred to new, labeled microcentrifuge tubes and stored at −80˚C until assayed. Plasma concentrations of corticosterone were determined by a commercial ELISA kit (EIA 4164, DRG Instruments GmbH, Marburg, Germany) in duplicates according to the manufacturers' instructions. Intra- and inter-assay coefficients of variation were below 10% and 12%, respectively.

## Oestrous cycle determination

The oestrous cycle stage was assessed by cytological analysis of vaginal smears to estimate the sex hormone status of the female mice. Vaginal smears were taken immediately after testing and postmortem after killing. Briefly, after each test, the female was placed on the cage lid with its hind end towards the experimenter. The rounded tip of a disposable pipette with 50 μl of sterile distilled water was gently placed at the opening of the vaginal canal, and vaginal smear cells were collected by lavage. Smears were placed on microscopic slides, allowed to dry, stained with 0.1% crystal violet solution, washed, and then analysed using light microscopy. The stage of the oestrous cycle was determined based on the relative ratio of nucleated epithelial cells, cornified squamous epithelial cells, and leukocytes. Since there were uneven distributions of oestrous cycle stages across groups on any given testing day [123], the vaginal smears data were combined into high-oestrogen state (proestrus, dioestrus-proestrus transition, and proestrus-oestrus transition) and low-oestrogen state (dioestrus, oestrus, metoestrus, oestrus-metoestrus transition, and metoestrus/dioestrus transition) and were included in the analysis as a linear binary factor.

## Statistical analysis of behavioural and physiological responses

All statistical analyses were performed using the statistical software R (version 3.6.2). A detailed R script is available as a supporting file. Data of male and female animals were analysed separately. Statistical tests and models employed for each analysis together with information on fixed and random factors are reported in S4–S9 Tables.

In brief, linear models without interaction terms and with identity link function were run for the body weight data collected at TP1 (i.e., right before killing within each RF; $n$ = 6 mice per rearing lab and sex; sample size was limited by the minimal number of cages with 3 mice per sex and RF) and for plasma corticosterone levels measured in the SRT ($n$ = 12 mice per RF and sex). Linear mixed models without interaction terms and with identity link function were run for body weight data and for relative adrenal weights ($n$ = 24 per RF and sex) collected at TP2.

Satterthwaite approximation was used for determination of $p$-values in the mixed models. A Dunn–Sidak Bonferroni correction method was applied to correct for multiple testing where necessary. For the physiological measures, such as body weight and plasma corticosterone levels, the threshold was set to $\alpha' = 1 - (1 - 0.05)^{1/3} = 0.0169$.

The distribution of the observed values for behavioural outcomes was inspected for deviations from normality with Q–Q plots. The data set of the physiological measures (body weights, relative adrenal weights, and corticosterone responses in the SRT) were normally distributed (S10–S12 Figs). Due to skewed distributions, transformations of behavioural data were necessary for 7 variables (S12 Fig). In the male cohort, OF latency, LDB latency, and LDB time in the light of males were square root transformed (S13A Fig). In the female cohort, OF distance was log transformed, OF time in the centre, OF latency, and LDB time in the light were square root transformed, while LDB latency was arcsine transformed (S13B Fig).

For the analysis of behavioural outcomes, we used a MANOVA without nesting and interaction terms and with identity link function. To check for correlation between recorded variables, we calculated Pearson's product moment correlation coefficient. We first excluded variables that were highly correlated (S12 Table), which resulted in the final list of 6 dependent behavioural variables (OF distance, OF time centre, OF latency, LDB time light, LDB entries into light, and LDB latency). Pillai's Trace was used to evaluate the MANOVA differences, while the robustness of the findings was confirmed by 3 other test statistics: the Wilk's Lambda, Hotelling–Lawley Trace, and Roy's Largest Root.

Multivariate outliers were identified by using the squared Mahalanobis distance (mvoutlier. CoDA version 2.0.9; [124]). The analysis suggested the existence of 6 outliers in the male cohort and 6 in the female cohort; S14 Fig), which were not removed for the further analysis. To investigate whether the outliers had an impact on the results, we rerun the analysis with the outliers removed and could confirm that results were not markedly different.

## ATAC-seq analysis on purified neuronal nuclei

**Nuclei isolation and fluorescence-activated nuclei sorting (FANS).** The ATAC-seq analysis was performed on ventral hippocampi isolated from male mice at TP1 (PND 56) and TP2 (PND 102). For this, test-naïve male mice were used to avoid effects of testing on the chromatin profile. Five cages per RF were selected randomly from all cages containing 3 male littermates after weaning. One mouse from the selected cages was killed in the RF at TP1 (total $n$ = 25, i.e., 5 biological replicates per rearing laboratory), while its test-naïve littermate was killed in the testing laboratory at the end of the experiment (TP2; total $n$ = 25, i.e., 5 biological replicates per RF).

The analysis has been restricted to males because they showed the most pronounced phenotypic differences in behaviour. Furthermore, in the female cohort, we would not be able to distinguish between the differences in chromatin organisation induced by hormonal fluctuations [125] and differences induced by differential rearing environments.

Total nuclei isolation and purification of neuronal nuclei were performed as described elsewhere [125,126] with slight modifications. In brief, the ventral hippocampus was dissected from one side of the brain at −20˚C, cut into small pieces, and stored in Eppendorf DNA LoBind 2 mL tubes at −80˚C until nuclei preparation. Nuclei preparation and sorting was performed in 5 batches per each TP, with each batch having exactly 1 sample from each facility in random order.

To obtain fresh nuclei, frozen tissue samples were resuspended in 4 ml of tissue lysis buffer (0.32 M Sucrose, 5 mM $CaCl_2$, 3 mM $Mg(CH3COO)_2$, 0.1 mM EDTA, 10 mM Tris-HCl (pH 8), 1 mM DTT, 0.1% Triton X-100) and dissociated by 30 strokes of pestle A (loose pestle) and then 20 strokes of pestle B (tight pestle) in a glass douncer (7 ml Dounce tissue grinder set, KIMBLE, DWK Life Sciences). The lysate was transferred to an ultracentrifuge tube, followed by adding 6 ml of sucrose buffer (1.8 M Sucrose, 3 mM $Mg(CH_3COO)_2$, 1 mM DTT, 10 mM Tris-HCl (pH 8)) underlaid beneath the solution. The samples were then spun at 171,192.8×$g$ in a Hitachi Ultracentrifuge (CP100NX; with Sorvall TH-641 swing bucket rotor) for 1 h at 4˚C. Next, the nuclei pellet was resuspended with 500 µl of 0.1% BSA in DPBS with glucose, sodium pyruvate, calcium, and magnesium. The nuclei solution was then incubated with monoclonal antibody against neuronal marker NeuN conjugated to AlexaFluor 488 (1:1,000; Merk Millipore, MAB377X) for 1 h at 4˚C on rotation protected from light. After incubation, DAPI (1:1,000; Thermo Fisher Scientific, 62248) was added to the reaction. The nuclei suspension was immediately taken to be processed on a FACSAria instrument (BD Biosciences, USA) at the Flow Cytometry and Cell Sorting Core Facility (FCCS CF) of the Department for BioMedical Research, University of Bern.

Prior to sorting, samples were filtered through a 35-μm cell strainer. To set up the gating protocol, we used 4 controls: (i) unstained nuclei only; (ii) DAPI only; (iii) IgG1 isotype control-AlexaFluor 488 and DAPI; and (iv) NeuN-AlexaFluor 488 only; in addition to a sample containing NeuN-AlexaFluor 488 and DAPI stain (S15 Fig), which allowed us to eliminate debris and any clumped nuclei effectively, resulting in an apparent separation of the NeuN + (neuronal) nuclei populations. From each individual ventral hippocampus, we collected 80,000 NeuN+ (neuronal) nuclei in BSA-precoated tubes filled with 200 μL of DPBS. The purity of the sorted nuclei was confirmed by resorting a small fraction of NeuN+ nuclei using the same protocol (showed more than 98% purity).

**Transposition reaction, ATAC-seq libraries preparation, and sequencing.** ATAC-seq was performed according to Buenrostro and colleagues [56], with some modifications. Following FANS, neuronal nuclei from the ventral hippocampus were spun down (2,900×$g$, 10 min at 4˚C). The supernatant was carefully removed, avoiding the visible nuclei pellet. The nuclei pellet was resuspended in 50 μl of the transposase reaction mix including 25 μL 2×TD reaction buffer and 3 μl Tn5 Transposase, (Illumina Tagment DNA TDE1 Enzyme and Buffer Kits, 2003419) and 22 μl of nuclease free water (NFW; Ambion, AM9937). The transposition reaction was performed at 37˚C for 30 min in a thermomixer with 500 RPM mixing, followed by purification using a MinElute PCR Purification Kit (Qiagen, 28004). The purified, transposed DNA was eluted in 10 μl of EB elution buffer and stored at −20˚C until amplification.

To amplify transposed DNA fragments, the following procedure was performed in 2 batches of 25 samples (1 batch per each TP) with equal group distribution ($n$ = 5 samples from each RF). For indexing and amplification of transposed DNA, we combined the following for each sample: 10 μl transposed DNA, 25 μl NEBNext High-Fidelity 2× PCR Master Mix (New England Biolabs, M0541S), 9 μl of unique, dual-indexed primer (IDT for Illumina Nextera DNA UD Indexes; 20026930), and 6 μl of NFW (Ambion, AM9937). The PCR reaction was carried out using the following conditions: 1 cycle of 72˚C for 5 min and 98˚C for 30 s, followed by 5 cycles of 98˚C for 10 s, 63˚C for 30 s, and 72˚C for 1 min, and a hold step at 4˚C.

We then performed a qPCR side reaction to manually assess the amplification profiles and determine the required number of additional PCR cycles [127]. The reaction mix was prepared by combining 5 μL of a previously PCR-amplified DNA with, 7.5 μl of SYBR Green PCR Master Mix (Applied Biosystems, 4344463), and 2.5 μl of NFW, and cycling conditions were set as follows: 1 cycle of 98˚C for 30 s, followed by 20 cycles of 98˚C for 10 s, 63˚C for 30 s, and 72˚C for 1 min. Under our experimental conditions, 2 to 4 PCR cycles were added to the initial set of 5 cycles. The amplified libraries were purified using MinElute PCR Purification Kit (Qiagen, 28004) and eluted in $20$ μL of the EB elution buffer. Library quality was monitored using the Advanced Analytics Fragment Analyzer CE12 (Agilent, USA; S15 Fig), and the concentration was determined by Qubit HS DNA kit (Life Technologies, Q32851) and quantitative PCR with the library quantification kit from Bioline Jet Set Library Quantification Kit LoROX (Meridian Bioscience, BIO-68029).

A total of 50 ATAC-seq libraries was sequenced in 2 batches with equal group distribution (25 libraries/batch/NovaSeq S1 Flow Cell; $n$ = 5 per each rearing laboratory) on the Illumina NovaSeq 6000 instrument with 2 × 100 bp pair-end protocol at the Next Generation Sequencing (NGS) Core Platform of the University of Bern.

**ATAC-seq data analysis.** Sequencing reads were trimmed and quality checked with fastp (version 0.20.1; [128]) with CTGTCTCTTATACACATCT as adapter sequence and a minimal read length of 30 bp. Reads were aligned to the mouse reference genome (ensembl build 102) with Bowtie2 in paired end mode (version 2.3.5.1; [129]) keeping only concordant and unique alignments.

Duplicate read pairs were marked using the *MarkDuplicates* command from the Picard software suite (version 1.140; broadinstitute.github.io/picard/). Peaks were then called in each

sample separately with MACS2 (version 2.1.4, with the parameters *-f BAMPE -g mm—nomodel -q 0.05—broad—broad-cutoff 0.1—keep-dup all*) as previously reported [130]. For each TP and RF, peaks were intersected with multovl (version 1.3; [131] and only peaks found in at least 3 samples per group (i.e., rearing laboratory) were kept (S16 Fig). Finally, peaks from all groups were merged with multovl (union of all peaks within groups). The number of reads within peak intervals was obtained with featureCounts (version 2.0.1; [132]) with the parameters—*primary—ignoreDup—minOverlap 30*. The peak set was further used to build a distance matrix using the aligned reads of the individual samples per each TP. The sample correlation matrices were generated for by using all sites and 10% of the most variable sites and visualised by correlation heatmaps, PCA, and t-SNE distance plots. Within each sample, the most and least accessible sites were defined as the peaks with the 2.5% highest and 2.5% lowest sequence counts. Sites from all samples and TPs were merged to generate the heatmap shown in Fig 3C.

The distance matrix was also used as an input for the PERMANOVA. By using PERMANOVA (function adonis() from the R-package vegan version 2.5–7 [133], we tested whether and to what extent the variation between samples can be explained by the RF within each TP. Since the test was based on 9,999 permutations, the lowest possible *p*-value was set to "<0.0001" instead of "0".

Variation in read counts was analysed with a general linear model in R (version 3.6.1) with the package DESeq2 (version 1.24.0; [134]) according to a factorial design with the 2 explanatory factors "RF" and "processing batch", within each TP. For the annotation of peaks, we used the ChIPseeker annotation for the plot with genomic features and the Homer annotation for TSS distance and candidate, protein coding, genes. Following specific conditions were compared with linear contrasts: (i) one-to-one (oto) comparison of each pair of laboratories (RF1 versus RF2, RF1 versus RF3, etc.) for each TP; (ii) one-to-many (otm) comparison of 1 laboratory to all other laboratories for each TP (RF1 versus all other RFs, RF2 versus all other RFs, etc.); and (iii) a global test for the factor "RF" (LRT_RF), i.e., do different rearing laboratories differ in general.

Within each comparison, *p*-values were adjusted for multiple testing (Benjamini–Hochberg), and regions with an adjusted *p*-value (false discovery rate (FDR)) below 0.01 and a minimal log2 fold-change (i.e., the difference between the log2-transformed, normalised sequence counts) of 0.5 were considered to be differentially accessible. Normalised sequence counts were calculated accordingly with DESeq2 and log2 (x + 1) transformed. GO and KEGG pathway analyses were performed on the significant peaks located 2 kb up- and downstream of the transcriptional start site. Functional annotation for enrichment of GO terms was performed using topGO (version 2.28; [135]) in conjunction with the GO annotation from Ensembl available through biomaRt [136]. Analysis was based on gene counts using the "weight" algorithm with Fisher's exact test (both implemented in topGO). Only GO terms with more than 5 genes were tested, and terms were identified as significant if the *p*-value was below 0.05. Enrichment of KEGG pathways in gene sets was tested with clusterProfiler (version 3.12.0; [137]) using the gene to pathway mappings available through biomaRt [136] and the package org.Rn.eg.db (version 3.8.2; [138]). Integrative Genome Viewer (IGV, version 2.8.9) was used to visualise and extract representative ATAC-seq tracks.

## Supporting information

**S1 Table. Detailed housing and husbandry conditions in each rearing facility (a) and testing laboratory (b).** https://doi.org/10.6084/m9.figshare.21088783.
(XLSX)

**S2 Table. ANOVA results for the effect of rearing facility (RF) and time point (TP) on α-diversity for males and females separately.**
(PDF)

**S3 Table. PERMANOVA partitioning variation in microbiome microbial community composition (β-diversity) between rearing facilities (RF), for each time point (TP) and sex separately.**
(PDF)

**S4 Table. Phenotypic variation in body weight of mice is induced by common differences between the rearing conditions in different facilities.** The effect of rearing environment on body weight was evaluated at 3 TPs throughout the study: right before killing within each RF (PND 56; TP1), after the acclimatisation period in the testing facility (PND 75), and at the end of the experiment (PND 102; TP2). Linear models were used to analyse data collected on mouse body weight at PND 56 (TP1) within each RF for males and females **(a)**. RF, litter size, and sex ratio at weaning were used as predictor variables. Linear mixed effect models with the same list of predictor variables as fixed effects were used for the body weight data collected in the testing facility for males and females **(b)**. Cage identification number (cage ID) in the testing facility was used as a random factor. (**a**) Linear regression model outcomes for body weight data collected in each RF; (**b**) Linear mixed effect model with type III ANOVA with Satterthwaite's approximation for body weight data collected at the testing facility. PND, postnatal day; RF, rearing facility; TP, time point.
(PDF)

**S5 Table. Phenotypic variation in behavior of mice is modified by common differences between the rearing conditions in different facilities.** MANOVA outcomes and statistics for males and females. The RF was defined as a main independent (predictor) variable. Additional covariates were included in the model based on the published evidence, which suggests that they might affect behavioral phenotype. List of the nuisance variables includes: litter size at weaning [1,2], sex ratio at weaning [3,4], and number of cage mates after weaning [5]. In addition, we included the stage of oestrous cycle at the time of testing (OF ECS and LDB ECS) in females, because of its effects on behaviour [6–8]. The results were confirmed by comparing the Pillai's Trace outcome with outcomes of 3 different test statistics. The proportion of variation in behaviour of mice, which is solely attributed to differences in rearing environments, was calculated by dividing the Pillai's trace by the degrees of freedom. MANOVA outcomes and statistics for behavioural parameters. LDB, light–dark box; MANOVA, multivariate analysis of variance; OF, open field; RF, rearing facility.
(PDF)

**S6 Table. Loadings of an LDA for male and female behaviour.** The rearing facility was defined as a grouping variable, while the 6 behavioral measures served as predictor variables. (OF = open field test, LDB = light–dark box test).
(PDF)

**S7 Table. Classification based on LDA of behaviour of males and females.**
(PDF)

**S8 Table. Phenotypic differences of the HPA stress profile in mice cannot be explained by common differences between the rearing conditions in different facilities.** Statistical outcomes of plasma corticosterone measures for males and females. We applied linear models, with the covariates rearing facility, litter size at weaning, sex ratio at weaning, and number of cage mates after weaning. In females, the stage of oestrous cycle on the testing day was also

included, and data were grouped into high- and low-oestrogen state.
(PDF)

**S9 Table. Phenotypic variation in relative adrenal weight of mice is induced by common differences between the rearing conditions in different facilities.** Statistical outcomes for relative adrenal gland weight data collected at the testing laboratory (TP1; PND 56) for males and females. Linear mixed effect models were used with rearing facility, litter size at weaning, sex ratio at weaning, and number of cage mates after weaning as fixed effect, while the cage ID in the testing facility was defined as a random factor. Linear mixed effect model with type III ANOVA with Satterthwaite's approximation for relative adrenal gland weight for males.
(PDF)

**S10 Table. PERMANOVA partitioning variation in chromatin accessibility profile between rearing facilities and processing batches. Manhattan distance between all samples was used as input**.
(PDF)

**S11 Table. The litter size, the litter sex ratio, and number of pups weaned from each litter across rearing facilities.**
(PDF)

**S12 Table. Pearson product moment correlations for behavioural measures for males (a) and females (b). Several highly correlated variables were removed before a multivariate analysis. Only variables with moderated correlations amongst each other were used. (a)** Pearson product moment correlations for behavioural measures for males. Mean absolute correlation coefficient for males was 0.20. (**b**) Pearson product moment correlations for behavioural measures for females. Mean absolute correlation coefficient for females: 0.29.
(PDF)

**S1 Fig. Timeline of the study.** BT mice, mice used for behavioural testing; GD, gestational day; LDB, light–dark box test; MA mice, mice that were not behaviourally tested and used for chromatin profiling and gut microbiome composition analysis; OF, open field test; PND, postnatal day; SRT,: stress reactivity test; TP, time point.
(PDF)

**S2 Fig. Diet suppliers: Differentially abundant taxa. (a)** Dendrogram representing relationship between samples from mice at TP1 following hierarchical clustering (average linkage). Horizontal bars below the dendrogram represent sample identity in relation to rearing facility (first bar) or diet supplier (second bar). (**b**) Mean abundance of amplicon sequence variants (ASVs) identified as differentially abundant between the 2 diet suppliers aggregated by phylum (left) and order (right). Significance symbols represent results from Wilcoxon paired-rank test. The raw data underlying this figure are available in the Figshare repository https://doi.org/10.6084/m9.figshare.21087688. The 16S rRNA gene sequencing data are available from the European Nucleotide Archive (ENA) under accession number PRJEB49361.
(PDF)

**S3 Fig. Oestrous cycle–dependent effects on behavioral phenotype in female mice from different rearing facilities.** Results of the open field (**a, b**) and light-dark box (**c**) tests are presented in females depending on the oestrous cycle stage determined immediately after behavioral testing. There was a significant effect of oestrogen status on time spent in the centre (**b**) and time spent in the light compartment (**c**) with high-oestrogenic females showing marginally higher activity than low-oestrogenic females. The raw data underlying this figure are

available in the Figshare repository https://doi.org/10.6084/m9.figshare.21087718.
(PDF)

**S4 Fig. Oestrous cycle–dependent effects on basal corticosterone levels in female mice from different rearing facilities.** Results are presented depending on the oestrous cycle stage determined after HPA reactivity tests. There was a significant effect of oestrogen status on basal corticosterone levels, with high-oestrogenic females showing lower basal corticosterone levels than low estrogenic females. The raw data underlying this figure are available in the Figshare repository https://doi.org/10.6084/m9.figshare.21087799.
(PDF)

**S5 Fig. Neuronal chromatin accessibility differs in males from different rearing facilities.**
(**a**) PCoA of the ATAC-seq data. (**b**) Manhatten distances between samples for closed chromatin sites visualised by t-SNE. (**c**) The number of significant differentially accessible peaks associated to each of rearing environments in a given TP (one-to-many comparison; adjusted for multiple testing FDR < 0.01 and abs(logFC) > 0.5. (**d**) The number of significant differentially accessible peaks between groups in a given TP (one-to-one comparison; adjusted for multiple testing FDR < 0.01 and abs(logFC) > 0.5). (**e**) Chromatin accessibility profiles of the Col19a1, Dlg 2, Fzd9, and Lrrc4c. Shown are genomic coordinates of differential ATAC-seq peaks. The raw data underlying this figure are available from the NCBI GEO database under accession number GSE191125. The analysis script is available at the GitHub repository https://github.com/MWSchmid/Jaric-et-al.-2022. ATAC-seq, assay for transposase-accessible chromatin using sequencing; Dlg 2, discs large homolog 2; FDR, false discovery rate; Fzd9, Frizzled9; GEO, Gene Expression Omnibus; Lrrc4c, Leucine-Rich Repeat-Containing 4C; PCoA, principal coordinate analysis; TP, time point.
(PDF)

**S6 Fig. Weaning strategy.** Each pup from a litter is sexed, weighed, and placed in a separate cage/container. After all animals from the litter are checked, it resulted in X number of separate cages/containers with males and Y number of separate cages/containers with females. If the number of males/females in a litter is 1, the animal is not weaned (female in Litter 1); if the number of males/females is 2 or 3, they are all weaned into same sex groups and taken to the housing room (Litter 2 and males in Litter 3); and if the number of males/females is >3, the 3 animals to be housed together are chosen using a random number generator (males in Litter 1 and females in Litter 3). Unweaned pups (female in Litter 1), extra pups, and dams were killed after the pups have been weaned.
(PDF)

**S7 Fig. Random down-sampling of 16S sequencing reads.** (**a**) Histogram of individual library sizes. (**b**) Example rarefaction plots of 8 randomly selected samples. The red line indicates rarefaction depth. The raw data underlying this figure are available in the Figshare repository https://doi.org/10.6084/m9.figshare.21087877. The 16S rRNA gene sequencing data are available from the European Nucleotide Archive (ENA) under accession number PRJEB49361.
(PDF)

**S8 Fig. Evaluation of differences in microbiome between samples from behaviorally tested BT mice and mice used for chromatin profiling MA mice.** (**a**) Mean values for α-diversity metrics for MA mice and BT mice. Top left: Chao1 richness; bottom left: observed species richness; top right: Pielou evenness, bottom right: Shannon diversity index. (**b**) Results of Wilcoxon signed-rank test for each α-diversity metric between MA and BT mice. (**c**) Ordination plot visualizing PCoA based on Bray–Curtis dissimilarity between samples from MA and BT

mice collected at TP2. (**d**) Result of PERMANOVA partitioning variation in microbiome composition between mice used for MA and BT. The raw data underlying this figure are available in the Figshare repository https://doi.org/10.6084/m9.figshare.21087931. The 16S rRNA gene sequencing data are available from the ENA under accession number PRJEB49361. ENA, European Nucleotide Archive; PCoA, principal coordinate analysis; PERMANOVA, permutational analysis of variance.
(PDF)

**S9 Fig. Power analysis.** Power curve for a one-way ANOVA with α = 0.05, k = 5 rearing facilities and *n* = 6 to 30 subjects, assuming an average means difference between 2 randomly chosen labs of 10% (blue), 20% (orange), or 30% (green). Power estimates are based on 10,000 repeated samples. To generate differences between labs we sampled distribution means from a normal distribution with the reported mean (10,000 square units) and standard deviation of 890, 1,780, and 2,260 square units, which resulted in samples, where the difference in the effect size between 2 randomly chosen labs was on average 10%, 20%, or 30%. The numerical data and code underlying this figure are available in the Figshare repository https://doi.org/10.6084/m9.figshare.21087931.
(PDF)

**S10 Fig. Q–Q (quantile–quantile) a probability plots for the body weight data points.** The data points of the body weights were normally distributed both in males (top panel) and females (bottom panel). The underlying numerical data are available in Fig 2 Data (Fig 2A MALES; Fig 2A FEMALES) in the Figshare repository https://doi.org/10.6084/m9.figshare.21081949.
(PDF)

**S11 Fig. Q–Q (quantile–quantile) a probability plots for the corticosterone response data points in the stress reactivity tests.** The data points of the corticosterone response were normally distributed both in males (top panel) and females (bottom panel). The underlying numerical data are available in Fig 2 Data (Fig 2C MALES; Fig 2C FEMALES) in the Figshare repository https://doi.org/10.6084/m9.figshare.21081949.
(PDF)

**S12 Fig. Q–Q (quantile–quantile) a probability plots for the relative adrenal gland weight at TP2.** The data points of the relative adrenal gland weights were normally distributed both in males (left plot) and females (right plot). The underlying numerical data are available in Fig 2 Data (Fig 2D MALES; Fig 2D FEMALES) in the Figshare repository https://doi.org/10.6084/m9.figshare.21081949.
(PDF)

**S13 Fig. Q–Q (quantile–quantile) a probability plots for the behavioral data sets.** Q–Q plots are presented before and after transformation of individual data points for both males (**a**) and (**b**) females. The underlying numerical data are available in Fig 2 Data (Fig 2B MALES; Fig 2B FEMALES) in the Figshare repository https://doi.org/10.6084/m9.figshare.21081949.
(PDF)

**S14 Fig. Multivariate outliers.** Results of the test for multivariate outliers indicated the existence of 3 outliers in males (**a**) and 3 outliers in the female data (**b**). Outliers are shown in red. The underlying numerical data are available in Fig 2 Data (Fig 2B MALES; Fig 2B FEMALES) in the Figshare repository https://doi.org/10.6084/m9.figshare.21081949.
(PDF)

**S15 Fig. Gating strategy for separation of neuronal nuclei using fluorescence-activated nuclei sorting (FANS).** Sorting plots from 3 negative controls (nuclei only, DAPI only, and Isotype control + DAPI) processed without primary antibody (neuron-specific marker NeuN), positive control containing NeuN antibody conjugated with Alexa 488 only (NeuN-Alexa 488 only control) and our sample processed with NeuN-Alexa 488 antibody and DAPI are shown. Representative FANS reports showing the gating strategy for the checking the size and granularity, removal of debris, and ensuring a successful separation of NeuN+ (neuronal) from non-neuronal single nuclei.
(PDF)

**S16 Fig. ATAC-seq library quality control.** The quality control of ATAC-seq libraries was performed by using Fragment Analyzer (FA). The representative FA trace with nucleosomal banding pattern is shown.
(PDF)

**S17 Fig. ATAC peak count statistics.** The number of peaks drops quite strongly with the minimal number of samples required for merging peaks. (**a**) Representative scatter plots showing log2 (x + 1) transformed, normalised values averaged in RF1 and RF2 at TP1. (**b**) Only high-confidence broad peaks, shared across at least 3 biological replicates of 1 group are used for all downstream analyses. The raw data underlying this figure are available from the NCBI Gene Expression Omnibus (GEO) database under accession number GSE191125. The analysis script is available at the GitHub repository https://github.com/MWSchmid/Jaric-et-al.-2022.
(PDF)

**S1 Data. Differentially abundant amplicon sequence variants (ASVs) between the 2 diet suppliers.** The 16S rRNA gene sequencing data are available from the European Nucleotide Archive (ENA) under accession number PRJEB49361. The supporting data are available at the Figshare repository https://doi.org/10.6084/m9.figshare.21088432.
(XLSX)

**S2 Data. Significant peaks and the gene annotations for TP1 (a) and TP2 (b).** The ATAC-seq data are available from the NCBI Gene Expression Omnibus (GEO) database under accession number GSE191125. The analysis script is available at the GitHub repository https://github.com/MWSchmid/Jaric-et-al.-2022. The supporting data are available at the Figshare repository https://doi.org/10.6084/m9.figshare.21088489.
(XLSX)

**S3 Data. Gene Ontology (GO). (a)** BP terms, which differ between rearing laboratories at TP1_one to one (oto) comparison; (**b**) BP terms, which differ between rearing laboratories at TP2_one to one (oto) comparison; (**c**) CC; terms that differ between rearing laboratories at TP1_one-to-one (oto) comparison; (**d**) CC; terms that differ between rearing laboratories at TP2_one-to-one (oto) comparison; (**e**) MF; terms that differ between rearing laboratories at TP1_one-to-one (oto) comparison; (**f**) MF; terms that differ between rearing laboratories at TP2_one-to-one (oto) comparison. The ATAC-seq data are available from the NCBI GEO database under accession number GSE191125. The analysis script is available at the GitHub repository https://github.com/MWSchmid/Jaric-et-al.-2022. The supporting data are available at the Figshare repository https://doi.org/10.6084/m9.figshare.21088504. ATAC-seq, assay for transposase-accessible chromatin using sequencing; BP, Biological Process; CC, cellular components; GEO, Gene Expression Omnibus; MF, molecular function.
(XLSX)

**S4 Data. KEGG pathways. (a)** KEGG pathways, which differ significantly in abundance between rearing facilities at TP1_one-to-one (oto) comparison; **(b)** KEGG pathways, which differ significantly in abundance between rearing facilities at TP2_one-to-one (oto) comparison. The ATAC-seq data are available from the NCBI GEO database under accession number GSE191125. The analysis script is available at the GitHub repository https://github.com/MWSchmid/Jaric-et-al.-2022. The supporting data are available at the Figshare repository https://doi.org/10.6084/m9.figshare.21088564. ATAC-seq, assay for transposase-accessible chromatin using sequencing; GEO, Gene Expression Omnibus; KEGG, Kyoto Encyclopedia of Genes and Genomes.
(XLSX)

**S1 Text. Sample size, Data exclusion, Replication, Randomisation, Blinding, ATAC-Seq Data deposition.**
(PDF)

## Acknowledgments

We would like to acknowledge the resources of the Flow Cytometry and Cell Sorting Core Facility (FCCS CF) of the Department for BioMedical Research and the Next Generation Sequencing (NGS) Core Platform of the University of Bern. Also, we would like to thank Dr. Stefan Müller for his assistance and support with nuclei sorting as well as Dr. Pamela Nicholson for assistance with Illumina sequencing. The authors thank Nadine Sudhof for excellent technical assistance and running the plasma corticosterone assays.

## Author Contributions

**Conceptualization:** Ivana Jaric, Bernhard Voelkl, Hanno Würbel.

**Data curation:** Ivana Jaric, Bernhard Voelkl, Melanie Clerc, Marc W. Schmid, Chadi Touma.

**Formal analysis:** Ivana Jaric, Bernhard Voelkl, Melanie Clerc, Marc W. Schmid.

**Funding acquisition:** Shinichi Sunagawa, Hanno Würbel.

**Investigation:** Ivana Jaric, Melanie Clerc, Janja Novak, Marianna Rosso, Reto Rufener, Chadi Touma.

**Methodology:** Ivana Jaric.

**Project administration:** Ivana Jaric.

**Resources:** Vanessa Tabea von Kortzfleisch, S. Helene Richter, Manuela Buettner, André Bleich, Irmgard Amrein, David P. Wolfer, Chadi Touma, Shinichi Sunagawa, Hanno Würbel.

**Supervision:** Shinichi Sunagawa, Hanno Würbel.

**Validation:** Ivana Jaric.

**Visualization:** Ivana Jaric, Melanie Clerc, Marc W. Schmid.

**Writing – original draft:** Ivana Jaric, Bernhard Voelkl, Hanno Würbel.

**Writing – review & editing:** Ivana Jaric, Bernhard Voelkl, Melanie Clerc, Marc W. Schmid, Janja Novak, Marianna Rosso, Reto Rufener, Vanessa Tabea von Kortzfleisch, S. Helene Richter, Manuela Buettner, André Bleich, Irmgard Amrein, David P. Wolfer, Chadi Touma, Shinichi Sunagawa, Hanno Würbel.

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
