## [Editor Report · Decision Letter 0]

31 May 2022

Dear Dr Jaric, 

Thank you for submitting your manuscript entitled "The rearing environment persistently modulates the phenotype of mice from the molecular to the behavioural level" for consideration as a Research Article by PLOS Biology.

Your manuscript has now been evaluated by the PLOS Biology editorial staff, as well as by an academic editor with relevant expertise, and I'm writing to let you know that we would like to send your submission out for external peer review.

Once your full submission is complete, your paper will undergo a series of checks in preparation for peer review. After your manuscript has passed the checks it will be sent out for review. To provide the metadata for your submission, please Login to Editorial Manager (https://www.editorialmanager.com/pbiology) within two working days, i.e. by Jun 02 2022 11:59PM.

Kind regards,

Roli Roberts

Roland Roberts

Senior Editor

PLOS Biology

rroberts@plos.org

---

## [Decision Letter · Decision Letter 1]

2 Aug 2022

Dear Dr Jaric,

Thank you for your patience while your manuscript "The rearing environment persistently modulates the phenotype of mice from the molecular to the behavioural level" went through peer-review at PLOS Biology. Your manuscript has now been evaluated by the PLOS Biology editors, an Academic Editor with relevant expertise, and by three independent reviewers.

You'll see that the reviewers are broadly positive about your study, but each raises a number of concerns that should be addressed by revision. During the course of these revisions, we ask that you provide at least some discussion of the points raised by R3 and how these issues may affect your conclusions.

In terms of more general editorial requests, we'd also ask that you make a few additional changes to your title and abstract. We'd suggest a title along the lines of "The rearing environment persistently modulates mouse phenotypes from the molecular to the behavioural level". We'd also suggest altering the concluding sentence of the Abstract to read: "Furthermore, they highlight an important limitation of inferences from single-laboratory studies and thus argue that study designs should take environmental background into account to increase the robustness and replicability of findings."

In light of the reviews, which you will find at the end of this email, we are pleased to offer you the opportunity to address the comments from the reviewers in a revision that we anticipate should not take you very long. We will then assess your revised manuscript and your response to the reviewers' comments with our Academic Editor aiming to avoid further rounds of peer-review, although might need to consult with the reviewers, depending on the nature of the revisions.

**IMPORTANT - SUBMITTING YOUR REVISION**

*Resubmission Checklist*

*Published Peer Review*

*PLOS Data Policy*

*Blot and Gel Data Policy*

Sincerely,

Roli Roberts

Roland Roberts, PhD

Senior Editor

PLOS Biology

rroberts@plos.org

REVIEWERS' COMMENTS:

Reviewer #1:

In the present study, the investigators utilized an innovative systematic multi-center approach to evaluate the effects of rearing environment on several behavioral and physiological endpoints in C57BL/6J mice. Briefly, the authors allocated and shipped pregnant mice from the same breeding stock to five independent animal facilities where their offspring were reared until eight weeks of age. After this period, all mice were transferred to a sixth and independent laboratory where they were evaluated on multiple endpoints of interest. This study design allowed the investigators to separate the impact of genetics and other environmental factors that can impact study outcomes at the time of testing to obtain a clearer understanding of the role of the rearing environment. There were many endpoints affected by rearing environment, for example a persistence in microbiome diversity differences, in addition to differences in body weights and behavioral phenotypes. Overall, this is a very nice study that provides some important information about variation between rearing environments and the limitations of studies originating from single laboratories. I only have a few comments/considerations listed below: 

1. The authors did a great job with their study design. I was especially impressed with the detour the driver took to ensure the travel time of the rats from the breeding facility to their new rearing environment was standardized. Just to confirm, all the pregnant mothers came from the same stock from Janvier Labs, does this mean that all mothers also came from the same colony room? I assume so, but this should be made clearer in the methods. For replication purposes, if animals came from multiple rooms at Janvier labs, it should be indicated which rooms, and how many dams from each room went to each rearing site. This is because there appear to be phenotypic differences across rooms in commercial facilities, not only across different breeding sites. Similarly, were all animals (dams and offspring) housed in the same colony room at each rearing site, or were males and females housed in different rooms, for example?

2. I may have misunderstood how it was written in the methods, but it seems there were at least some cases where multiple same-sex pups from a single litter were used on a measure. If so, how were litter effects controlled for? The authors should make a table showing how many litters (and how many total of each male and female offspring) from each of the five rearing environments were represented on each endpoint of interest. 

3. The authors report that variation in litter size did not impact behavior. Litter size was not standardized either within or across rearing environments. The authors should provide a table showing the range of litter sizes across each of the rearing environments. Were they comparable across sites? Litter size prior to weaning can impact later life body weight, and this could affect differences between stress physiology (indeed basal cort was different as a function of litter size), microbiome, and behavior. Can the authors test if litter size acts as a moderating variable between body weight and the other endpoints of interest?

4. Gut microbiome changes have a known influence on social behavior and pain sensitivity. Did the authors look at any behavioral tests evaluating these, or other tests evaluating cognitive measures?

5. Supplementary Tables 10a and 10b are placed towards the end of the supplementary materials. However, these tables contain many important methodological details that answered several questions I initially had. I suggest these two tables be moved from supplementary and into the main paper.

Reviewer #2:

[identifies herself as Vanessa Beijamini]

Jaric and colleagues investigated the influence of common differences in the rearing environment on behavior, gut microbial community, and epigenetic of male and female adult inbred mice. They proposed an important hypothesis and an interesting experimental design to test it, controlling the mice genotype and test conditions. The work is clearly and accurately presented, with sufficient details of methods and results. The experimental design is appropriated to test the hypothesis. The conclusions drawn were adequately supported by the results.

Notwithstanding, the authors followed very good practices to increase reproducibility (detailed methods, study pre-registration, sample size calculation, randomization, blindness of experimenters, provision of additional data, and detailed statistical analysis). 

Minor Issues:

1. Why were data from male and female mice analyzed separately?

2. In the light-dark box, was there a delay between releasing the animal into the dark compartment and opening the door to avoid entry into the light compartment due to escape motivation from the experimenter? 

Reviewer #3:

In this manuscript, the authors described a comprehensive study on how environmental differences between animal facilities affect mouse phenotypes. The authors randomly allocated pregnant mice from the same breeding cohort to five different facilities and analyzed the phenotypes of progenies before and after transportation to a single test laboratory. They found the gut microbiomes had different compositions and heterogeneity, liked caused by the differences in diets. Mice from different facilities also showed different body weights and behavioral phenotypes. They mapped the chromatin accessibility in the ventral hippocampus by ATAC-seq and detected differences in genes involved in synaptic plasticity, neurogenesis and signaling pathways. The authors proposed that animal studies should be done across facilities to increase the robustness and replicability of findings.

Please find below several questions and suggestions that I hope will improve the manuscript.

1. The authors reported several phenotypic and molecular differences in mice from different facilities. It would be very helpful if the authors could study or discuss whether these phenotypic and molecular differences are connected to help reveal the underlying mechanisms. For example, does the difference in microbiome composition drive the difference in weight gain and behavior phenotypes? The authors could transfer fecal contents of mice from different facilities into germ-free mice and see if the microbiota causally causes differences in weight, behavior, or neural chromatin structures. The authors may cohouse the mice from different facilities or treat them with antibiotics to see if the gut microbiome drives these phenotypical and molecular differences. 

2. The authors suggested that the difference in diet drives the difference in the gut microbiome. The authors may use two different diets to confirm whether diet drives phenotypic and molecular differences when mice are housed in the same facility. A related question is that about 20% of variances in microbiome and behaviors can be explained by the difference in animal facilities. While the authors listed some other factors such as litter size, litter sex ratio and group size, how do these factors increase/affect explainable variances? The result will help evaluate the significance of animal facilities in driving phenotypic variation.

3. In Fig 3c, there are some big changes in chromatin accessibility between TP1 and TP2. What are the genes that are regulated by these peaks? Technically, it is unclear how the authors defined the most and least accessible sites.

4. In several places, figure references are mislabeled. For example, line 182 should be Fig 2c instead of fig 1c. Line 185, where are Fig1 g,h. On line 219, where is Fig. 2g? Should be a supplemental figure.

5. PND should be defined in figure 2 legends.

---

## [Editor Report · Decision Letter 2]

7 Sep 2022

Dear Dr Jaric,

Thank you for your patience while we considered your revised manuscript "The rearing environment persistently modulates mouse phenotypes from the molecular to the behavioural level" for publication as a Meta-Research Article at PLOS Biology. This revised version of your manuscript has been evaluated by the PLOS Biology editors and the Academic Editor.

Based on the Academic Editor's assessment of your revision, we are likely to accept this manuscript for publication, provided you satisfactorily address the following data and other policy-related requests.

IMPORTANT: Please attend to the following:

a) We note that in the previous round, one of the reviewers requested that you move a supplementary table (Table S10 in the original submission) into the main paper. In your rebuttal you state that you have done this; however, it is still in the supplement (as Table S1, I think). Please move it into the main paper (as Table 1).

b) Please provide a blurb, according to the instructions in the submission form.

c) Please address my Data Policy requests below; specifically, we need you to supply the numerical values underlying Figs 1CDEFGH, 2ABCD, 3ABCDEFG, S2AB, S3ABC, S4, S5ABCDE, S7AB, S8ABCD, S9, S10, S11, S12, S13AB, S14AB, S15, S16, S17, either as a supplementary data file or as a permanent DOI’d deposition like Zenodo, Dryad, Figshare, etc.

d) Please cite the location of the data clearly in all relevant main and supplementary Figure legends, e.g. “The data underlying this Figure can be found in S1 Data" or “The data underlying this Figure can be found in https://zenodo.org/record/XXXXX”

We expect to receive your revised manuscript within two weeks. 

*Published Peer Review History*

*Press*

Sincerely,

Roli Roberts

Roland Roberts, PhD

Senior Editor,

rroberts@plos.org,

PLOS Biology

DATA POLICY:

Regardless of the method selected, please ensure that you provide the individual numerical values that underlie the summary data displayed in the following figure panels as they are essential for readers to assess your analysis and to reproduce it: Figs 1CDEFGH, 2ABCD, 3ABCDEFG, S2AB, S3ABC, S4, S5ABCDE, S7AB, S8ABCD, S9, S10, S11, S12, S13AB, S14AB, S15, S16, S17. NOTE: the numerical data provided should include all replicates AND the way in which the plotted mean and errors were derived (it should not present only the mean/average values).

SPECIES INDICATED IN THE ABSTRACT? 

- Please note that per journal policy, the model system/species studied should be clearly stated in the abstract of your manuscript. 

DATA NOT SHOWN?

---

## [Editor Report · Decision Letter 3]

20 Sep 2022

Dear Dr Jaric,

Thank you for the submission of your revised Meta-Research Article "The rearing environment persistently modulates mouse phenotypes from the molecular to the behavioural level" for publication in PLOS Biology. On behalf of my colleagues and the Academic Editor, Cilene Lino de Oliveira, I'm pleased to say that we can in principle accept your manuscript for publication, provided you address any remaining formatting and reporting issues. These will be detailed in an email you should receive within 2-3 business days from our colleagues in the journal operations team; no action is required from you until then. Please note that we will not be able to formally accept your manuscript and schedule it for publication until you have completed any requested changes.

Sincerely, 

Roli Roberts

Senior Editor

PLOS Biology

rroberts@plos.org